# Cross-shore parallel tidal channel systems formed by alongshore currents

Zeng Zhou [1,2], Yizhang Wei[2], Liang Geng [1,3] ✉, Ying Zhang[2,4], Yuxian Gu[2], Alvise Finotello [5], Andrea D'Alpaos [5], Zheng Gong[1], Fan Xu [3] ✉, Changkuan Zhang[2] & Giovanni Coco[6]

Parallel tidal channel systems, characterized by commonly cross-shore orientation and regular spacing, represent a distinct class of tidal channel networks in coastal environments worldwide. Intriguingly, these cross-shore oriented channel systems can develop in environments dominated by alongshore tidal currents, for which the mechanisms remain elusive. Here, we combine remote sensing imagery analysis and morphodynamic simulations to demonstrate that the deflection of alongshore tidal currents at transitions in bed elevation determines the characteristic orientation of the parallel tidal channels. Numerical results reveal that sharp changes in bed elevation lead to nearly 90-degree intersection angles, while smoother transitions in bed profiles result in less perpendicular channel alignments. These findings shed light on the potential manipulation of tidal channel patterns in coastal wetlands, thus equipping coastal managers with a broader range of strategies for the sustainable management of these vital ecosystems in the face of climate change and sea level rise.

Tidal channel systems, carved by the rhythmic ebb and flow of the tide across mudflats and wetlands along the world's coastlines, serve as vital drainage conduits within these ecosystems[1]. These channel systems facilitate essential processes such as sediment transport and water quality regulation, and provide habitat for diverse species[2], thereby bolstering biodiversity and ecosystem resilience[3,4]. Their importance extends to mitigating storm surge impacts[5] and contributing to carbon sequestration, highlighting their role in climate change adaptation and global carbon cycling. Studying these channel systems is critical for understanding coastal ecosystem dynamics, guiding conservation efforts, and informing coastal management strategies to ensure the sustainability and resilience of these critical habitats against environmental change.

Recognized for their diversity and heterogeneity, tidal channel systems embody the interplay of a variety of biogeophysical forces across varied coastal settings[6]. Decades of research have been devoted to dissecting the intricate morphologies that define these channel systems, with emphasis on their chaotic branching patterns and meandering extensions[7–22]. However, amidst this endeavor to map their complexity, a specific subset of tidal channel systems, distinguished by orderly growth patterns and distribution, has often been overlooked. These systems display parallel branches that extend nearly straight at angles around 90° to the shoreline or the thalweg of their originating channels (Fig. 1, see also Table S1.1 in Supplementary Information). Even though the channel branches in different systems exhibit distinct degrees of curvature, which is dominated by local flow strength and bed friction[8,9] (also introduced in Section 1 of the Supplementary Information), the shape and trend of each branch in the same parallel tidal channel system are consistent. This exceptional uniformity has prompted some researchers

[1]The National Key Laboratory of Water Disaster Prevention, Hohai University, Nanjing 210024, China. [2]Jiangsu Key Laboratory of Coastal Ocean Resources Development and Environment Security, Hohai University, Nanjing 210024, China. [3]State Key Laboratory of Estuarine and Coastal Research, East China Normal University, Shanghai 200062, China. [4]Bureau of Water Resources of Luhe District, Nanjing 211500, China. [5]Department of Geosciences, University of Padova, Padova, Italy. [6]Faculty of Science, University of Auckland, Private Bag 92019, Auckland, New Zealand. ✉e-mail: gengliang1991@hhu.edu.cn; fxu@sklec.ecnu.edu.cn

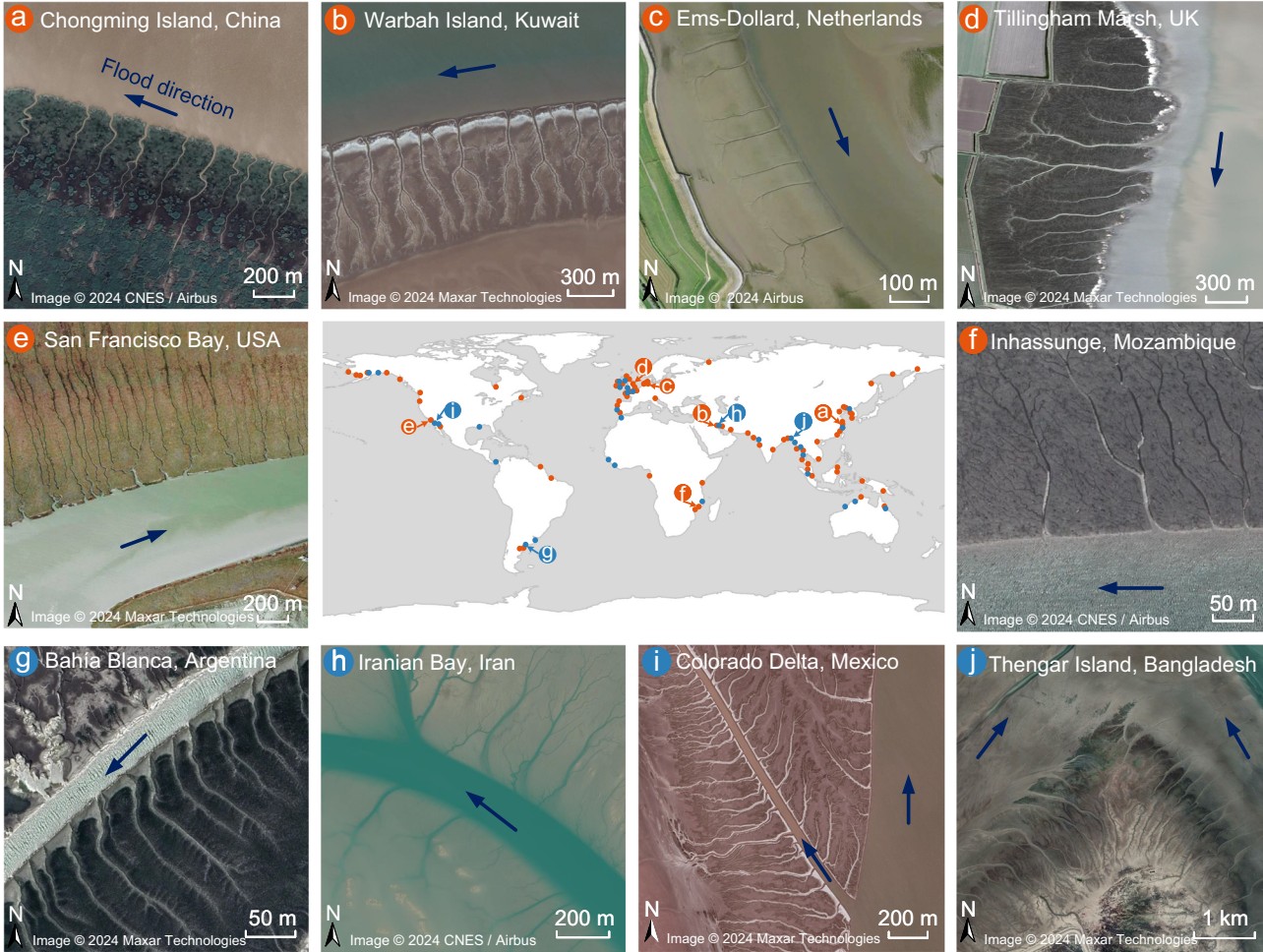

**Fig. 1 | Global distribution of parallel channel systems with a few examples.** **a–f** Examples of linear parallel branches that are nearly straight and generally perpendicular to the parent channel. **a** Chongming Island, China, Image © 2024 CNES/Airbus. **b** Warbah Island, Kuwait, Image © 2024 Maxar Technologies. **c** Ems-Dollard, Netherlands, Image © 2024 Airbus. **d** Tillingham Marsh, UK, Image © 2024 Maxar Technologies. **e** San Francisco Bay, USA, Image © 2024 Maxar Technologies. **f** Inhassunge, Mozambique, Image © 2024 CNES/Airbus. **g–j** Examples of parallel branches that are not very straight and are oblique to the parent channel. **g** Bahía Blanca, Argentina, Image © 2024 Maxar Technologies. **h** Iranian Bay, Iran, Image © 2024 CNES/Airbus. **i** Colorado Delta, Mexico, Image © 2024 Maxar Technologies. **j** Thengar Island, Bangladesh, Image © 2024 Maxar Technologies.

to delineate them as a separate entity, aptly termed the parallel tidal channel system.

Parallel drainage patterns observed in terrestrial landscapes often occur on steeper slopes, where water flows rapidly downhill, parallel to the slope's contour[23,24]. However, the geomorphology and dynamics of parallel channel systems in coastal environments are markedly different: they are normally found in low-gradient landscapes such as mudflats and marsh surfaces[4,25], where the erosion processes are largely controlled by water-surface gradients rather than topographic ones[8,9,11,26], and intriguingly, their dynamics are dominated by alongshore tidal flows, whose strength and water level change periodically, yet their orientation is cross-shore. Some studies speculate that the formation of parallel channel systems may represent a stable biogeomorphic state coexisting with a uniform distribution of vegetation, in contrast to a more complex stable state characterized by sinuous channels and mixed vegetation patterns[27,28]. Conversely, other research argues that parallel channel systems are transient morphological features shaped by equal-strength tides and periodic storms acting on coastal lines[2]. However, these past investigations of parallel tidal channels have been largely descriptive and have not concretely elucidated the relationship between their forms and the seemingly contradictory coastal flow dynamics[1,12,27].

To fill this knowledge gap, we conducted comprehensive morphological analyses of 275 parallel channels selected from 21 sites worldwide, encompassing diverse geomorphological settings, including estuaries, lagoons, and open coasts (background information about the selected areas can be found in Sections 1 & 2 of Supplementary Information). Moreover, to gain deeper insights into the physical processes underlying parallel channel inception, we performed morphodynamic simulations to disentangle the influence of alongshore tidal currents and bed topography. Our study serves as a pioneering effort to unravel the processes governing the formation and evolution of parallel channel systems in coastal environments, bridging the knowledge gap and expanding our understanding of tidal channel dynamics.

## Results

### Branching angles of parallel channel systems

The defining characteristic of parallel channel systems is the consistent orientation of individual channel branches, which can be quantified by the angle they form with either the parent channel or the shoreline. Since the direction of individual channel branches can change significantly when moving landward from the intersecting point (i.e., branch root) toward the tip of the branch (i.e., branch head), we defined two distinct angles to comprehensively describe channel

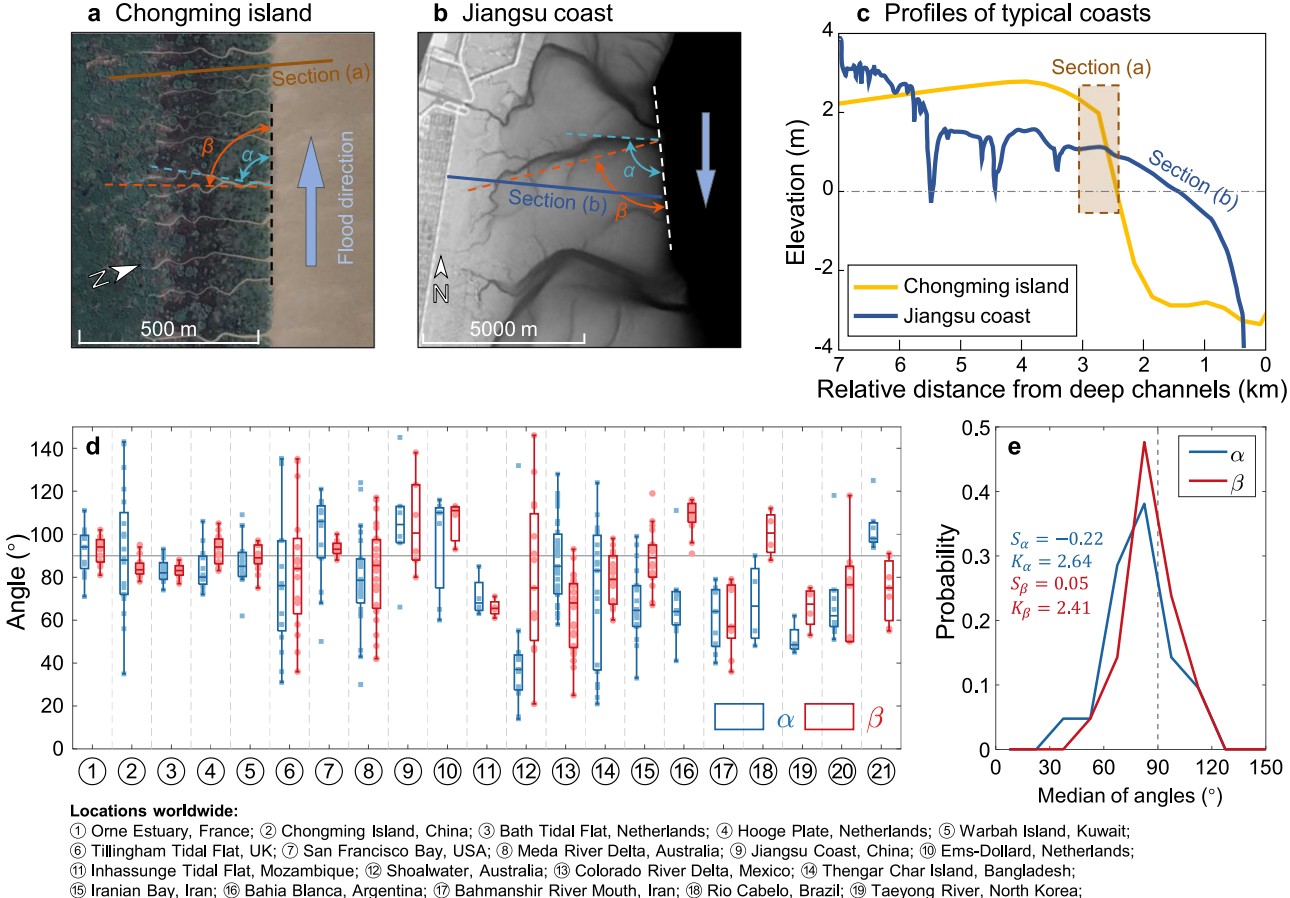

**Fig. 2 | Channel angles of typical parallel channel systems. a, b** The channel systems formed in two typical tidal flat environments with the definition of channel angles marked. **a** Chongming Island, China, Image © 2024 CNES/Airbus. **b** Jiangsu Coast, China, adapted from LiDAR survey data (2006). **c** The cross-sectional profiles of the two typical tidal flat environments. **d** Comparisons between connecting channel angles ($\alpha$) and overall channel angles ($\beta$) of natural parallel channel systems. Box plots indicate median (middle line), 25th, 75th percentile (box) as well as the minimum and maximum values which are not an outlier (whiskers). The scattered points represent the original angle of each channel. **e** The probability distribution of the median of channel angles for 21 selected regions. $S_\alpha$ and $S_\beta$ express the skewness of the distribution, which is a measure of the asymmetry of the data around the sample mean, with the positive value signifying the data spreads out more to the right of the mean than to the left, while the negative value signifying the data spreads out more to the left. $K_\alpha$ and $K_\beta$ express the kurtosis of the distribution, which is a measure of how outlier-prone a distribution is.

branch morphology. The first one is the intersection angle $\alpha$ [°] measured at the branch root, while the second one is the trend angle $\beta$ [°] that defines the overall orientation of the entire branch from head to root (Fig. 2a, b and Fig. S1.1 in Supplementary Material file). To maintain a standardized measurement procedure, we invariably measured the $\alpha$ and $\beta$ angles based on aerial imagery captured at low tide, using the direction of flood flows as a reference (see Fig. 2a, b).

The overall distribution of channel angles and the probability distribution of the median of the angles for all the 21 selected regions are shown in Fig. 2d, e. Different from distributary channel networks found in river deltas, whose mean bifurcation angle is typically close to 72°[22], our analysis reveals that the mean intersection angle, $\alpha$, and mean trend angle, $\beta$, of all the selected channel systems are 78.9° and 83.9°, respectively. The variability in channel angles, with a large portion of empirical data showcasing values smaller than 90° (Fig. 2e), suggests a potential deflection of near-shore flows toward the dominant along-shore flow direction, whether it be ebb or flood. However, some parallel tidal channel systems show regular patterns with angles approaching 90° (i.e., Fig. 1a–f). The higher degree of morphological variability we observed across our study sites likely results from local variations in tidal hydrodynamics, landforms, vegetation distribution, and combinations thereof. For instance, both the Colorado River Delta in Baja California, Mexico, and Thengar Island in Bangladesh are

situated within estuaries. This location aids in determining flood directions from satellite images by examining the estuary's shape. Within these settings (refer to Fig. 1i, j), parallel channel systems are shaped by alongshore tidal currents originating from two distinct parent channels, leading to generally smaller channel angles overall. Notably though, even in the case of parallel channels, variance in the intersection angles is more pronounced than in the channel trend angles (Fig. 2d), and the probability distribution of the median of intersection angles has a smaller peakedness and more significant skewed shape (Fig. 2e), again suggesting larger morphological variability near the shoreline.

As depicted in Fig. 1a–f, parallel channel systems exhibiting regular patterns emerge on intertidal flat areas bordering parent channels characterized by steep banks. We thus contend that tidal flat topography serves as the principal contributor to the observed variability in channel angles. Please note that in some coastal areas established by marshes and affected by wave erosion, steep cliffs also present along the marsh edges. However, in such scenarios, tidal channels might not usually develop. Therefore, the wave-erosion-dominated marshes are not the focus of this paper. To analyze the effects of tidal flat topography shaped by alongshore currents, two typical tidal channel systems are compared in Fig. 2a, b. The first tidal basin is located in the north branch of the Changjiang estuary (China), with a topographic

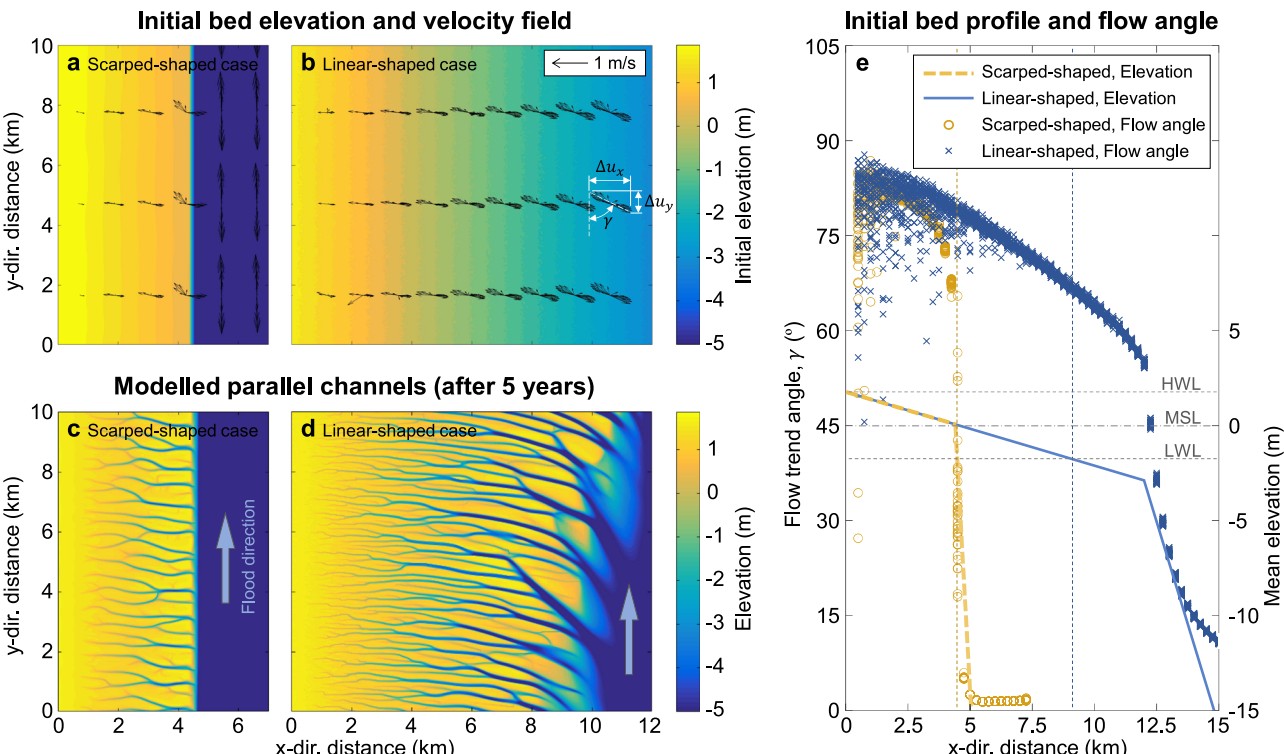

**Fig. 3 | Numerical modeling of parallel tidal channel systems. a, b** The initial bed elevation and velocity variations of the two numerical models. **c, d** The modeled tidal channel systems after 5 years. **e** The two initial bed profiles and the mean flow trend angle of the two cases. The blue and yellow vertical dashed lines denote the lower boundaries of the intertidal zone of the two cases.

profile characterized by a sharp transition in elevation (i.e., knickpoint) around the mean sea level, e.g., between the estuary and the mudflat platform, the latter being sub-horizontal (see yellow line in Fig. 2c). In this context, the primary morphodynamic process at play is the persistent alongshore tidal current within the estuary, which gives rise to steep escarpments along the channel banks. Through the cyclical influence of tides, the formation of parallel channels begins at the escarpment knickpoint (highlighted by a brown box in Fig. 2c), with channels maintaining a perpendicular alignment to the main parent channel (as illustrated in Fig. 2a). However, in contrast to the sharp change in bed topography observed in Chongming Island, the selected tidal basin in the Jiangsu coast (China) features an approximately uniform seaward bed slope. This characteristic results in a much wider intertidal zone, as illustrated by the blue line in Fig. 2c, and leads to distinct parallel channels that are oblique relative to the shoreline, especially close to the branch roots (Fig. 2b).

These observations highlight the complex interplay of ecomorphodynamic processes that contribute to the morphological diversity observed even within apparently similar parallel tidal channel systems. To disentangle the factors responsible for the observed morphological variability, we conducted numerical simulations using a custom morphodynamic model (see Section 3 of the Supplementary Material). The model allowed us to systematically vary parameters related to alongshore tidal currents, tidal range, and bed slope, enabling the examination of their individual and combined effects on the variability of parallel channel angles.

**Cross-shore channel morphology and alongshore flow**
The above-mentioned observations suggest that the topographic profile of channel banks plays a critical role in dictating the pattern of parallel channel systems. To test this hypothesis, we first design two morphodynamic modeling experiments featuring a periodically submerged tidal platform adjacent to a deep channel (the detailed model

description and setup are introduced in Fig. S3.1 of the Supplementary Material). These two simulated tidal basins are affected by the alongshore currents with a tidal range of 3.5 m and consist of an upper gently sloping tidal flat plain and a lower steep slope. The main difference between the two is the position of the inflection point where the topographic profile transitions from a gentle to steep slope (the yellow dashed line and blue solid line in Fig. 3e). For the first case, the inflection point is topographically higher and corresponds to the mean sea level, leading to an abrupt topographic change within the intertidal zone. This case will be referred to as the "Scarped-shaped case". For the second case, the inflection point lies below the minimum tide level, resulting in a uniformly sloping intertidal zone, which will be referred to as the "Linear-shaped case" hereinafter.

We used an open-source hydro-morphodynamic model, Delft3D, to simulate the formation of the parallel channel systems. The model successfully captures the formation and evolution of parallel channel systems formed by alongshore tidal flows (detailed comparisons of the channel patterns between the model and the reality can be found in Section 4 of the Supplementary Information). The flood currents come from the north and gradually change their direction as they enter the tidal flat platform. During the ebb phase, the water on the platform drains into the offshore domain and flows to the south (see Fig. 3a, b). Randomly imposed initial bed perturbations (magnitude scale of 0.1 m) promote flow concentration and further trigger unevenly distributed channels on tidal flats. As the adjacent flow paths merge, small channel branches intersect and form a tree-like structure. After 5 years, the modeled channel systems exhibit morphological patterns similar to those of natural systems, with overall trend angles $\beta$ being approximately equal to 90° (Fig. 3c, d). In particular, both intersection angles $\alpha$ and trend angles $\beta$ in the abruptly changing topography (Scarped-shaped case) are close to 90° (Fig. 3c), whereas the intertidal zone with uniform slope produces intersection angle $\alpha$ generally smaller than 90° (Fig. 3d).

The different profile shape characteristics observed in the two numerical simulations indicate that an abrupt change in bed slope significantly affects the morphological development of parallel channel systems (Fig. 3c, d). Furthermore, we try to explain this impact from a hydrodynamic perspective. Figure 3a, b shows the field of tidal currents during the early evolution stage with two different starting profile shapes. As shown in Fig. 3a, b, the water flow exhibits the characteristics of reversing currents, and its velocity ellipse appears elongated, indicating that the effect of tidal currents at any location of the profile is mainly concentrated in one direction. To characterize the direction of this trend, we calculated the flow trend angle at any location, which can be measured as:

$$\tan \gamma = \frac{|\Delta u_x|}{|\Delta u_y|} \tag{1}$$

where $\Delta u_x$ and $\Delta u_y$ represent the changes in flow velocity during a tidal cycle (e.g., Fig. 3b). Therefore, $\gamma$ approximately shows the influence trend of the currents on the direction of channel branches, and it has a variation range between 0° and 90°. A low value of $\gamma$ indicates a predominant $y$-directional (i.e., longshore) component in tidal flow. Conversely, as $\gamma$ approaches 90°, it suggests that the $x$-directional, cross-shore flow component is significantly stronger than the alongshore component.

Due to the continuity of tidal currents, the alongshore flow shifts to cross-shore to feed and drain the inner area, leading to landward-increasing values of flow trend angle ($\gamma$) and overall channel angle ($\beta$). In the Linear-shaped case (blue cross symbols in Fig. 3e), the intertidal zone is wide enough for the alongshore ebb and/or flood currents to adjust their direction. Additionally, as the ebb currents gradually turn to flood currents at low tides, the flow smoothly transitions between 0° and 90°. Conversely, in the Scarped-shaped case, the sharp change in bed slope results in a steep bank that restricts the $x$-directional component of the tidal current when the water level is lower than the inflection point. Consequently, tidal currents experience a fast diversion as they flow over the inflection point on the profile, whose elevation is near the mean sea level. This diversion in flow direction occurs quickly because of the steep rise in the tide at this moment, leading to a relatively large channel angle at the roots of the parallel branches. It is also notable that in the lower intertidal zone, especially at the inflection points of the Scarped-shaped case, the flood tides tend to veer toward the north (i.e., parallel to the shoreline), while the ebb tides exhibit a tendency perpendicular to the shoreline (as shown by the velocity ellipses in Fig. 3a, b). This indicates that the flood currents provide a stronger contribution to generating channel branches with small intersection angles ($\alpha$), while the joint contribution of ebb currents and topographic gradients carves more perpendicular channels. Overall, the bed profile shape, characterized by the profile slope and local elevation gradient, plays an important role in the parallel channel pattern formation.

### Role of tidal range, bed slope, and local topographic relief

Based on the simulation of the tidal flats under the effect of the alongshore currents, we found that it is easier to generate a parallel tidal channel system that is uniform in size and consistent in morphology when there is a sudden change in the bed slope. However, other environmental factors may have an impact on the morphology of the parallel tidal channel systems. Therefore, we carried out several more simulations to explore the sensitivity of channel patterns and channel angles to other factors, such as tidal range, slope of upper platform, and the magnitude of local topographic reliefs (Fig. 4). Consistently with the channel systems observed through remote sensing (Fig. 2e), all the simulated channels are perpendicular to the parent channel, with trend angles ($\beta$) being approximately equal 90°. It

indicates that the alongshore currents turn to the cross-shore direction when traveling in shallow waters, no matter what the tidal environment is. In contrast, a smaller value and a higher variability are observed for the intersection angles ($\alpha$) as shown in Fig. 4d, which agree with the more dispersed distribution of the median of intersection angle in remote sensing statistics (Fig. 2e), indicating that the pattern of the parallel branch root is more easily affected by the alongshore currents.

Tidal range is found to play a key role in determining the planform of parallel channel systems (Fig. 4a). In cases where the tidal range is small, the size and cross-shore extent of the tidal channels formed are also limited. Additionally, weaker hydrodynamics make the influence of terrain on the tidal channel morphology more pronounced, and the morphology of the tidal channels tends to be more perpendicular to the shoreline. Compared with the case of a small tidal range, a larger tidal range (6 m) results in a smaller intersection angle ($\alpha$), suggesting that the stronger tidal currents have more significant influences on the roots of parallel branches, and generate non-perpendicular parallel channel systems similar to those shown in Fig. 1h.

The case of a horizontal upper platform (yellow lines in Fig. 4b) has the smallest overall and intersecting channel angles due to the weaker variation in bed elevation and frictional resistance compared to other cases. As a result, the alongshore currents can maintain relatively stronger energy and shape parallel branches that feature smaller $\alpha$ and $\beta$ angles. Notably, in the case with steeper platforms (blue dashed lines in Fig. 4b), the channel angles are also quite small and the parallel channel branches are typically short and develop only in the proximity of the shoreline. This is because the larger bed slope reduces the size of the intertidal zone and diminishes the area that permits alongshore currents to transition into cross-shore currents. Consequently, the formation of perpendicular parallel tidal channels requires the intertidal platform to meet a suitable slope range.

By increasing the magnitude of initial local topographic relief, the influence of terrain on tidal channel systems is enhanced. On the one hand, increased local topographic relief leads to a greater spread in the overall angles of the tidal channels. The variable topography tends to generate tidal channel systems of varying sizes and uneven distribution (Fig. 4c). On the other hand, increasing local topographic relief also exhibits a similar effect as reducing tidal range, such as an increase in the intersection angle is found in the cases with tidal range of 2 m and local topographic relief magnitude of 1 m. This also indicates that the relative strength of terrain and hydrodynamics is a key factor in controlling the morphology of parallel channel systems.

## Discussion

In agreement with the observation of various natural systems worldwide, our model results demonstrate that parallel tidal channels can be formed by a combination of alongshore tidal forcing and alongshore uniformity in bed topography (e.g., similar bed profile shape and local topographic relief magnitude in the alongshore direction). The creeks present on fringing tidal flats originate at the transition zone between the milder-slope upper and steeper-slope lower flats where the highest ebb velocities and largest velocity gradients tend to occur[29]. The parent channel provides a conduit for the alongshore flow constrained within the banks, and the lateral gradient of bed elevation controls the bend of the flow direction. Therefore, parallel tidal channels usually occur as "first order" branches of a parent channel, while for sub-branches, the local bed geometry, local topographic relief, and flow field are often not uniform alongshore and thus do not meet the prerequisite of the formation of parallel sub-channels. In fact, individual parallel channels may further branch into lower order creeks, preferentially with dendritic patterns, because the condition of the alongshore uniformity is not satisfied anymore. The alongshore uniformity of the bed topography and the magnitude of local topographic relief may also influence the drainage processes on the tidal basin and the spacing of channel systems, which need further analysis in the future.

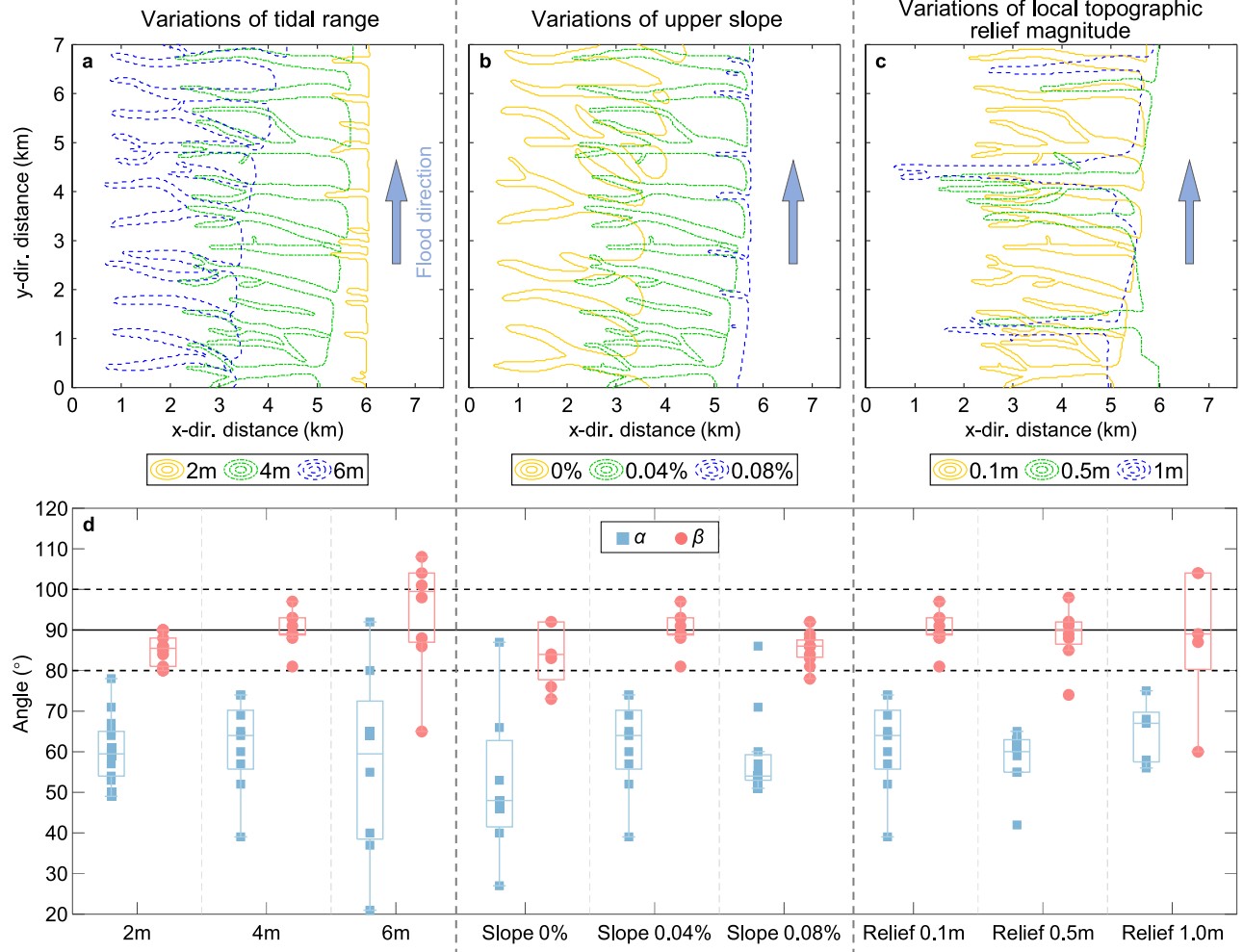

**Fig. 4 | The sensitivity of channel patterns and angles to environmental factors.** **a**–**c** The plane profiles of channel systems generated in environments with different tidal ranges, upper slopes, and local topographic relief magnitudes. **d** The corresponding overall channel angles (red marks) and connecting channel angles (blue marks) are plotted in the panel. Box plots indicate median (middle line), 25th, 75th percentile (box) as well as the minimum and maximum values which are not an outlier (whiskers). The scattered points represent the original angle of each channel.

The characteristic tidal flow field over tidal networks and flats is friction-dominated and can be approximated by a Poisson-type equation deduced from the shallow water equation[9]. In this view, the bending of the streamline can be attributed to the lateral water-surface gradient around the transitional area between the channel and the flat, so as to balance the sudden increase of the frictional resistance (see Supplementary Material file, Fig. S5.1). Field observations also documented the process of changing flow direction as the tide spreads between the tidal channel and the tidal platform[30–32] (see Figs. S3.2 and S3.3 in Section 3 of Supplementary Information). With the gradual rise of the water level, there is a transition of flow direction from alongshore to cross-shore. Furthermore, field observations carried out in the gently sloping tidal flat on the Jiangsu Coast (China), indicate that the alongshore component of the velocity ellipse gradually decreases moving landward, while the cross-shore component increases[32,33]. Conversely, in the north branch of Changjiang River, the velocity is mainly oriented along the channel, due to the flow restriction imposed by the steep channel banks[34].

Further in-depth and systematic research is needed to enhance our understanding of parallel tidal channel systems. Numerous factors, such as vegetation growth rates and colonization strategies, sediment physical properties, and human-induced changes in topography, may influence channel morphology. Moreover, detailed investigations are needed to examine aspects such as the characteristic spacing of channel branches, drainage efficiency, sinuosity development over

time, and possible channel bifurcations. To advance research in this field, it is essential to collect more comprehensive field data, which should encompass tidal flows, sediment properties, and high-resolution digital elevation data.

The systematic study of parallel tidal channels can provide strong support for coastal management and ecological environment protection in the future. First of all, since the parallel tidal channel system is controlled by hydrodynamic and topographic conditions, managers can judge the local environmental conditions and changes based on the morphology of parallel channels through remote sensing, such as the tidal current regime and ecosystem states[35], providing help for the overall management of tidal flats, especially in the context of global climate change such as sea level rise. Second, it also provides a way to artificially shape the tidal channel system: by adjusting the strength of the alongshore current and the surrounding terrain, the secondary tidal channel system can be shaped as needed to meet the local demand for water, sand, and material exchanges.

## Methods
### Remote sensing image processing
We extracted tidal channel networks from multispectral satellite imagery, enabling a comprehensive comparison of parallel channel characteristics worldwide. To gain a representative understanding of these characteristics on a global scale, we selected 275 parallel channels from

21 sites across various geomorphological settings, including open coast, lagoon, and estuary environments (Table S1.1). For each channel, we quantified the degree of parallelism by measuring two angles formed with the parent channel or shoreline. The first one is the intersection angle $\alpha$ [°], measured at the branch root, while the second one is the trend angle $\beta$ [°], defining the overall orientation of the channel from head to root (Fig. 2a, b and Fig. S1.1). Here, to ensure consistent comparison across tidal systems with varying orientations, we measured these two angles using the direction of flood flows as a reference.

### Numerical modeling

A hydro-morphodynamic modeling package, Delft3D, is used to study the formation of parallel channel systems (further details about the model can be found in Section 3 of Supplementary Information). The model first simulates water levels and flow velocities by solving the depth-averaged shallow water equations (Table S3.1). Hydrodynamic results are then used to calculate sediment transport of both cohesive (i.e., mud) and non-cohesive (i.e., sand) sediment fluxes. The deposition and erosion fluxes of cohesive sediment are calculated by an advection–diffusion equation with the Partheniades–Krone formulations[36]. With regard to the transport of non-cohesive sediment, the formulation of Soulsby–van Rijn[37] is adopted, which calculates the total sediment transport as a summation of bed load and suspended load (Table S3.1). The calculated sediment transport of both cohesive and non-cohesive sediments results in a morphological change that is updated every hydrodynamic time step. The new bed level is then utilized to compute the flow field at the next hydrodynamic time step, thereby completing the morphodynamic loop[13,38]. To reduce the computational cost of morphological models, a morphological factor approach is applied that has been widely examined by numerous morphodynamic simulations[39,40].

Parallel tidal channels typically develop on open-coast tidal flats or along the flanks of major tidal channels, forced by alongshore tidal currents (Fig. 1, Table S4.1, and Fig. S4.1). Here we simulate two tidal flats with distinct data-based topographies to examine the formation of two different parallel channel systems observed in China. The model domain consists of an initially unchanneled tidal flat and a deeper offshore area (Fig. S3.1). The initial bed elevation at the landward end of tidal flats is set to 1.8 m above mean sea level, decreasing linearly to 0 m or −3 m depending on various topographies of the two cases, while the offshore area attains a sharp slope from 0 m or −3 m near the shore to −20 m at the offshore boundary (Fig. S3.1). This large depth prevents shallowing due to sedimentation outside the intertidal areas[41]. The grid resolution is set to 25 m by 25 m to enable the evolution of channel networks.

This model covers the eastern, southern, and northern sea boundaries (Fig. S3.1). At the eastern sea boundary, we apply an M2 tidal cycle with a range of 3.5 m, as observed in field surveys (Table S2.1). The northern- and southern-seaward boundaries are set as Neumann conditions with zero water-level gradients. Additionally, a phase difference of 17.24 deg/h is set to generate alongshore currents[42], mimicking a tidal flow environment along one side of major tidal channels.

The model incorporates both mud and sand transport, for which the sediment parametrizations are set similar to field observations[43] (Table S3.2). The initial bed composition comprises a 10-m well-mixed sediment fraction, with 5 m of mud and 5 m of sand. At the inflow boundaries, we employ the concept of equilibrium sediment concentration, ensuring that sediments entering through sea boundaries adapt to local flow conditions. This approach maintains stability of the model boundary, with the sediment concentration gradient perpendicular to the open boundary equal to 0[44]. To simplify the model, no sediment inputs are considered at the open boundaries.

### Reporting summary

Further information on research design is available in the Nature Portfolio Reporting Summary linked to this article.

## Data availability

The data regarding the worldwide parallel channels are available as Supplementary Material (Supplementary Data 1).

## Code availability

The model Delft3D is an open-source code available online (at https://oss.deltares.nl). The model setup is available at the repository Zenodo (https://zenodo.org/records/10814687).

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

## Acknowledgements

Z.Z. acknowledges funding from the National Key R&D Program of China (2022YFC3106201), the National Natural Science Foundation of China (42361144873, 42376161), the Carbon Peak & Carbon Neutral Science and Technology Innovation Project of Jiangsu Province (BK20220020), and the Fundamental Research Funds for the Central Universities (B230201061). L.G., F.X., and Z.G. acknowledge funding from the National Natural Science Foundation of China (42206162, 42376168, 51925905). L.G. further acknowledges funding from the China Post-doctoral Science Foundation (Grant 2022M711019). We thank Prof. Zhuojia Fu and Xiaotian Zhang from Hohai University for their helpful discussion and valuable assistance.

## Author contributions

Z.Z. and Y.Z. conceived of the study. Z.Z., L.G., and F.X developed the theory, designed the model, and wrote the manuscript with input from A.F., G.C., A.D., Y.W., and Y.G. Y.W., Y.G., and Y.Z. collected and digitized the satellite imagery of tidal channel networks. Y.W. and Y.G. worked out the numerical simulations. Z.G. and C.Z. supervised the project.

## Competing interests

The authors declare no competing interests.
