## [Peer Review File · Nature Communications]

Cross-shore parallel tidal channel systems formed by alongshore currentsREVIEWER COMMENTS

Reviewer #1 (Remarks to the Author):

This manuscript addresses the unresolved question of what processes create straight tidal channels. While much attention has been paid to the geomorphology of sinuous tidal channels, comparably little has been paid to that of straight tidal channels. The authors address this research gap by conducting a geographically widespread remote sensing analysis of marshes that exhibit the parallel channel geomorphology and developing a morphodynamic model that they use to test the hypothesis that relatively uniform topography and a steep escarpment can lead to the formation of parallel channels, which their results uphold. The paper is very well written and interesting, and the figures are compelling. The work is worthy of a broad audience. However, there are several areas in which it falls short of my expectations:

1. It does not contextualize the work in previous literature discussions of parallel tidal channels. For example, parallel and sinuous channels were recognized as alternative stable states by Moffett et al. (2015; full references below) and Moffett and Gorelick (2016). Subsequently, Larsen (2019) proposed mechanisms through which parallel tidal channels could develop, including through erosional processes acting on relatively uniform initial topography, as tested here. However, additional mechanisms were also suggested that are dependent on vegetation type (whereby clumpy vegetation may favor parallel channels) and the initial conditions from which the channels are established and whether conditions are erosional or depositional. Additionally, there is a missed opportunity to discuss the findings in the context of channel pattern research in fluvial geomorphology, wherein it is well understood that steep terrain favors the development of parallel channels, whereas less steep terrain favors the development of sinuous channels.
2. Within the context of the above literature discussion, the omission of a specific role for vegetation or heterogeneous roughness in the model seems to exclude critical processes in the geomorphic development of tidal marshes. While I appreciated the sensitivity analysis conducted in the model, varying the critical shear stress parameter (through which vegetation could implicitly be represented) spatially or across runs did not seem to have been done. It is important to at least discuss the putative role of vegetation and how not explicitly simulating it might have impacted the results.
3. I was unsatisfied with the lack of discussion of how preexisting conditions could influence the development of tidal channels, which I believe could be key to understanding these dynamics. For example, steep escarpments are unlikely to be places where tidal channels would newly form, as these are typically found in places where marshes are established but subject to wave erosion. It does not seem likely to me, then, that the initial topography selected for the model would be representative of where tidal channels would be newly forming.
4. The 80-100-degree demarcation between parallel and oblique channels seems somewhat arbitrary. Is there a justification? It seems that the more important counterfactual against which to test the

mechanism for the development of parallel channels would be the development of sinuous channels. There is a strong precedent in fluvial geomorphology for contrasting processes that produce straight versus sinuous channels.

5. The connection between the remote sensing component of the project and the modeling was a bit weak, and it seems that opportunities to test the hypothesis through observational data were missed (see comments below on Recommendations for Future Work).

6. There is no statement of data/code availability, and there is not enough data/information for the work to be reproduced.

MODERATE AND MINOR COMMENTS

- The model is mentioned only very briefly outside the supplemental section. I suggest including at least a paragraph describing the model in the main text, as details about what the model does and does not include are essential to the interpretation of the results.
- Even in Supplementary Section 3, the model description does not contain an adequate level of detail, including description of how the equations were solved (e.g., via finite element or finite difference) and on what platform, what the grid resolution and time step were, and what the boundary conditions were for water and sediment. Is the code openly available to readers? I would encourage the authors to embrace best practices for open, reproducible science in making data and code available.
- Line 104-105: “data dispersion of intersection angles is more pronounced than in the trend angles.” I had to read this phrase many times before it made sense to me. Please rephrase. Maybe “Variance in the intersection angles is more pronounced than in the channel trend angles.”
- Figure 3: The subfigures are mislabeled in the caption and in the text references.
- Line 178: Larger compared to what?
- Line 224-226: This statement appears to be driven by one result. I’m not convinced that it is an accurate/representative statement.
- Figure 4a: I suggest omitting the TR in the legend of figure 4a, as the abbreviation is not defined and it is not consistent with other subfigures.
- Paragraph starting on line 261: Much of this text is repetitive of the first paragraph in this section.
- Line 282-284: The statement about relevance to management is vague. I suggest deleting it or making it more specific.

RECOMMENDATIONS FOR FUTURE WORK

If the authors are looking to extend their analysis in their future work, it would be interesting to see whether varying spatial features of an imposed topography (such as a wavelength, if the initial topography were to be initialized nonrandomly) would influence the outcome, given that multipatch flow-sediment feedbacks are strongly dependent on the spacing of roughness elements.

Another interesting element to consider in future work might be extracting digital elevation model data from the marshes analyzed through remote sensing. A powerful test of the authors' hypothesis might be whether the topographic characteristics associated with parallel vs. oblique channels in the observational data agree with the conclusions from modeling.

REFERENCES CITED

Larsen, L. G. Multiscale flow-vegetation-sediment feedbacks in low-gradient landscapes. *Geomorphology* 334, 165–193 (2019).

Moffett, K. et al. Multiple Stable States and Catastrophic Shifts in Coastal Wetlands: Progress, Challenges, and Opportunities in Validating Theory Using Remote Sensing and Other Methods. *Remote Sensing* 7, 10184–10226 (2015).

Moffett, K. B. & Gorelick, S. M. Alternative stable states of tidal marsh vegetation patterns and channel complexity. *Ecohydrology* 9, 1639–1662 (2016).

Reviewer #2 (Remarks to the Author):

This is an interesting paper dealing with connectivity geometry between secondary channels and the adjacent water body (trunk channel, bays, open water, etc.) in estuarine environments. They provide examples from around the world to demonstrate applicability of their study. As stated their major objective was to “disentangle the influence of alongshore tidal currents and bed topography” in the formation of these systems.

Figure 2 is difficult to read and understand and is of poor quality. The aeriels showing the channels are small and difficult to see. Panel 2d is hard read too. The aeriels I Figure 1, c, h, & j are of poor quality as well.

Line 102: “For instance, parallel channels in the Colorado River Delta (Baja California, Mexico) and Thengar Island (Bangladesh) are influenced by alongshore tidal currents from two different parent channels, resulting in overall smaller channel angles (see Fig.1i, j).” As this is a remote sensing study, how are authors able to characterize the tidal flow? There are no citations. On what basis are they able to make that observation?

Line 111: “By comparing the topography of tidal flats from the satellite images (Fig. 1), we deem that the variations in tidal flat topography are the main causes for the formation of two types of parallel channels.” It has been shown that the erodibility of the substrate is an important parameter in affecting

channel evolution, and you present no data on the sedimentology (grain size, shear strength, etc.) of these systems. Moreover, the comparisons between the channel profiles in Figure 2 are confusing, as the Jiangsu channel also appears to have a knickpoint but less dramatic.

Line 181. This statement is highly significant because it their explanation of why the two systems are different: “On the contrary, because of the sharp change of bed slope in the Scarped-shaped case, the steep bank restricts the x-directional component of the tidal current when the water level is lower than the inflection point. The tidal currents then experience a fast diversion as they flow over the inflection point on the profile, leading to a relatively large channel angle at the roots of the parallel branches.” However, this needs to be much better stated. In the linear case, the tide gradually reverses flow (from ebb to flood) as the tide rises, because of the gradual slope. However, in the scarped case water doesn’t enter the secondary channels until later in the tidal cycle when the tide is rising steeply thereby producing a rapid change in current direction and strong flood currents. I would also guess that the longshore current reaches a maximum after the flow significantly diminishes in the secondary channel in the linear case. In the scarped case, the longshore current is maximum after the water level has dropped below the secondary channel opening. Again, in paper, this is not well explained.

The strength and timing of the longshore current is highly germane to this paper, and we are provided with no information of its character (time-velocity asymmetry).

Larger tidal ranges lead to greater tidal prisms and flow of the longshore current thereby increasing the tendency of longshore transport, spit formation and deflection of the secondary channel mouth.

A major deficiency of this paper is the lack of discussion of how tidal flats versus marsh systems behave.

Reply to Reviewers

Note to the Reviewers:

The comments and suggestions of the reviewers are copied in normal font. The reply to each comment by the reviewers is written in blue font and appears just after the original comment or question. The modified/added sentences have been copied from the revised manuscript for the convenience of the reviewers.

We wish to thank the reviewers for their valuable and constructive comments that have certainly resulted in a much more insightful manuscript.

Reviewer Comments

Reviewer #1 (Remarks to the Author):

This manuscript addresses the unresolved question of what processes create straight tidal channels. While much attention has been paid to the geomorphology of sinuous tidal channels, comparably little has been paid to that of straight tidal channels. The authors address this research gap by conducting a geographically widespread remote sensing analysis of marshes that exhibit the parallel channel geomorphology and developing a morphodynamic model that they use to test the hypothesis that relatively uniform topography and a steep escarpment can lead to the formation of parallel channels, which their results uphold. The paper is very well written and interesting, and the figures are compelling. The work is worthy of a broad audience. However, there are several areas in which it falls short of my expectations.

Reply: We thank the reviewer for the positive remarks and constructive comments. The issues raised are critical and relevant to improve the quality of the manuscript, and have been all carefully considered and addressed.

1. It does not contextualize the work in previous literature discussions of parallel tidal channels. For example, parallel and sinuous channels were recognized as alternative stable states by Moffett et al. (2015; full references below) and Moffett and Gorelick (2016). Subsequently, Larsen (2019) proposed mechanisms through which parallel tidal

channels could develop, including through erosional processes acting on relatively uniform initial topography, as tested here. However, additional mechanisms were also suggested that are dependent on vegetation type (whereby clumpy vegetation may favor parallel channels) and the initial conditions from which the channels are established and whether conditions are erosional or depositional. Additionally, there is a missed opportunity to discuss the findings in the context of channel pattern research in fluvial geomorphology, wherein it is well understood that steep terrain favors the development of parallel channels, whereas less steep terrain favors the development of sinuous channels.

Reply: We agree with the reviewer that a more comprehensive review of previous studies was needed. Accordingly, we have rewritten the Introduction section and added the previously missing but important literature such as *Moffett et al. (2015)*, *Moffett and Gorelick (2016)* and *Larsen (2019)*. In the revised introduction section (lines 36-88), we first introduce the importance of studying tidal channel networks. Secondly, we emphasize the specificity of the parallel tidal channel system and point out that there are few relevant studies. Then we briefly review the previous studies on the formation of parallel drainage systems in terrestrial and coastal environments. We highlight the critical role of the topographic gradient on the formation of parallel channels found in terrestrial drainage systems, and touch upon parallel vs. sinuous channels developing under the influence of vegetation, as suggested. In the end, we briefly describe our research methods and objectives. The rewritten introduction can be found in the revised manuscript (lines 36-88), and reads:

“Tidal channel networks, carved by the rhythmic ebb and flow of the tide across mudflats and wetlands along the world’s coastlines, serve as vital drainage conduits within these ecosystems¹. These networks facilitate essential processes such as sediment transport and water quality regulation, and provide habitat for diverse species², thereby bolstering biodiversity and ecosystem resilience^{3,4}. Their importance extends to mitigating storm surge impacts⁵ and contributing to carbon sequestration, highlighting their role in climate change adaptation and global carbon cycling. Studying these networks is critical for understanding coastal ecosystem dynamics, guiding conservation efforts, and informing coastal management strategies to ensure the sustainability and resilience of these critical habitats against environmental change.

Recognized for their diversity and heterogeneity, tidal channel networks embody the interplay of a variety of biogeophysical forces across varied coastal settings⁶. Decades of research have been devoted to dissecting the intricate morphologies that define these networks, with particular emphasis on their chaotic branching patterns and meandering extensions⁷⁻²². However, amidst this endeavor to map their complexity, a specific subset of tidal channel systems, distinguished by orderly growth patterns and distribution, has often been overlooked. These systems display parallel branches that extend nearly straight at angles around 90 degrees to the shoreline or the *thalweg* of

their originating channels (Fig.1, see also Table S1.1 in supplemental material file). Even though the channel branches in different systems exhibit distinct degrees of curvature, which is dominated by local flow strength and bed friction^{8,9} (also introduced in Section 1 of the supplemental material file), the shape and trend of each branch in the same parallel tidal channel system are consistent. This exceptional uniformity has prompted some researchers to delineate them as a separate entity, aptly termed the Parallel Tidal Channel System.

Parallel drainage patterns observed in terrestrial landscapes often occur on steeper slopes, where water flows rapidly downhill, parallel to the slope's contour^{23,24}. However, the geomorphology and dynamics of parallel channel systems in coastal environments are markedly different: they are normally found in low-gradient landscapes such as mudflats and marsh surfaces^{4,25}, where the erosion processes are largely controlled by water-surface gradients rather than topographic ones^{8,9,11,26}, and intriguingly, their dynamics are dominated by alongshore tidal flows, whose strength and water level change periodically, yet their orientation is cross-shore. Some studies speculate that the formation of parallel channel systems may represent a stable biogeomorphic state coexisting with a uniform distribution of vegetation, in contrast to a more complex stable state characterized by sinuous channels and mixed vegetation patterns^{27,28}. Conversely, other research argues that parallel channel systems are transient morphological features shaped by equal-strength tides and periodic storms acting on coastal lines². However, these past investigations of parallel tidal channels have been largely descriptive and have not concretely elucidated the relationship between their forms and the seemingly contradictory coastal flow dynamics^{1,12,27}.

To fill this knowledge gap, we conducted comprehensive morphological analyses of 275 parallel channels selected from 21 sites worldwide, encompassing diverse geomorphological settings, including estuaries, lagoons, and open coasts (Background information about the selected areas can be found in Section 1&2 of Supplementary Material). Moreover, to gain deeper insights into the physical processes underlying parallel channel inception, we performed morphodynamic simulations to disentangle the influence of alongshore tidal currents and bed topography. Our study serves as a pioneering effort to unravel the processes governing the formation and evolution of parallel channel systems in coastal environments, bridging the knowledge gap and expanding our understanding of tidal channel dynamics.”

2. Within the context of the above literature discussion, the omission of a specific role for vegetation or heterogeneous roughness in the model seems to exclude critical processes in the geomorphic development of tidal marshes. While I appreciated the sensitivity analysis conducted in the model, varying the critical shear stress parameter (through which vegetation could implicitly be represented) spatially or across runs did not seem to have been done. It is important to at least discuss the putative role of vegetation and how not explicitly simulating it might have impacted the results.

Reply: We agree with the reviewer that the presence of vegetation and the corresponding heterogeneous roughness should have been considered to investigate

channel development and final channel patterns. Moreover, we recognize the importance of systematically and comprehensively discussing the impact of vegetation on parallel tidal channels.

To address this aspect, we have incorporated new simulations to mimic the co-evolution of channels and vegetation. This approach has enhanced our understanding of vegetation effects on channel dynamics. These simulations were carried out following the methods outlined in the Bio-geomorphological model by *Best et al. (2018)*, which accounts for the growth of vegetation and its effects on bed roughness and drag coefficient.

Taking the case of a scarped-shaped intertidal plain as an example (as shown in Fig. S1), we conducted a five-years-long simulation and observed that the presence of vegetation promotes the landward extension of channels, leading to the formation of a greater number of channel branches, which are, on average, longer compared to simulations without vegetation. This consequently increases drainage density, as defined by *Marani et al. (2003)*. These findings align with previous studies (e.g., *Kearney et al., 2016, Van de Vijzel et al., 2023, Geng et al., 2023*) suggesting that vegetation increases drainage density and efficiency in intertidal plains dissected by tidal channel networks. However, it is important to note that simulations involving the presence of vegetation also result in the formation of parallel channel systems similar to those obtained when running simulations without vegetation, especially in terms of channel junction angles. While this similarity may stem from model limitations, such as its inability to simulate vegetation-driven channel meandering (a feature common to all tidal network morphodynamics models with the exception of *Mariotti and Finotello, 2023*), we believe that the effects of vegetation still require comprehensive exploration and specific analyses that are beyond the aim of the present work.

Figure S1. Comparison of bed topography after one-year and five-years evolution in Scarp-shaped case with and without vegetation.

Additionally, we wish to emphasize that parallel channels can form in coastal environments even in the absence of vegetation. For example, refer to Fig. S2, selected from Fig. 1c,h in the main text, which illustrates this point. This indicates that vegetation presence is not necessary for the formation of parallel channels. In certain coastal areas, parallel channels may form before vegetation colonization occurs, and subsequently established vegetation may strengthen the pre-existing channels channel, enhancing their stability and promoting second-order planform development such as lateral migration and meandering. However, in situations where vegetation is established prior to or simultaneously with development, the effects of vegetation on parallel channels remain unclear and should be carefully addressed in future studies. We have included a paragraph in the discussion section to elaborate on potential avenues for future research (please see our response below on Recommendations for Future Work).

Figure S2. Illustrative instances of parallel channels observed in exposed mud flats, exemplified by those found in Ems-Dollard, Netherlands (a), and the Iranian Bay, Iran (b), are presented. These satellite images are additionally featured in Fig. 1c&h within the main text. Source: Google Earth.

Marani, M., Belluco, E., D'Alpaos, A., Defina, A., Lanzoni, S., & Rinaldo, A. (2003). On the drainage density of tidal networks. Water Resources Research, 39(2).

Best, Ü. S., Van der Wegen, M., Dijkstra, J., Willemsen, P. W. J. M., Borsje, B. W., & Roelvink, D. J. (2018). Do salt marshes survive sea level rise? Modelling wave action, morphodynamics and vegetation dynamics. Environmental modelling & software, 109, 152-166.

Kearney, W. S., & Fagherazzi, S. (2016). Salt marsh vegetation promotes efficient tidal channel networks. Nature communications, 7(1), 12287.

Van de Vijzel, R. C., van Belzen, J., Bouma, T. J., van der Wal, D., Borsje, B. W., Temmerman, S., ... & van de Koppel, J. (2023). Vegetation controls on channel network complexity in coastal wetlands. Nature communications, 14(1), 7158.

Geng, L., Lanzoni, S., D'Alpaos, A., Sgarabotto, A., & Gong, Z. (2023). The Sensitivity of Tidal Channel Systems to Initial Bed Conditions, Vegetation, and Tidal Asymmetry. Journal of Geophysical Research: Earth Surface, 128(3), e2022JF006929.

Mariotti, G., & Finotello, A. (2023). A flow-curvature-based model for channel meandering in tidal marshes. ESSOAR Open Archive.

3. I was unsatisfied with the lack of discussion of how preexisting conditions could influence the development of tidal channels, which I believe could be key to understanding these dynamics. For example, steep escarpments are unlikely to be places where tidal channels would newly form, as these are typically found in places where marshes are established but subject to wave erosion. It does not seem likely to me, then, that the initial topography selected for the model would be representative of where tidal

channels would be newly forming.

Reply: We appreciate the reviewer's comment regarding the influence of preexisting conditions on the development of tidal channels, as it is indeed a crucial aspect in understanding these dynamics.

In our study, we focused on the formation of parallel tidal channels dominated by alongshore tidal currents, with preexisting conditions mainly referring to the bed topography (including steep escarpments or linear slopes). Our observations from satellite images (see Fig. 1) confirm the presence of steep escarpments occurring at the outer bank of large channel bends. These escarpments result from sustained lateral erosion at channel outer banks, leading to the formation of steep outer banks where parallel tidal channels indeed form (Fig. 1). This is the type of pre-existing conditions we have tried to generate and test.

We agree with the reviewer that the steep escarpments are often associated with marshes subject to wave erosion, where tidal channels may not typically form. In order to clarify the specific coastal environment considered in this study, we briefly introduced the differences between the two environments in lines 129-136:

“As depicted in Fig. 1a-f, parallel channel systems exhibiting regular patterns emerge on intertidal flat areas bordering parent channels characterized by steep banks. We thus contend that tidal flat topography serves as the principal contributor to the observed variability in channel angles. Please note that in some coastal areas established by marshes and affected by wave erosion, steep cliffs also present along the marsh edges. However, in such scenarios, tidal channels might not usually develop. Therefore, the wave-erosion-dominated marshes are not the focus of this paper. To analyze the effects of tidal flat topography shaped by alongshore currents, two typical tidal channel systems are compared in Figure 2a-b.”

We have also emphasized the dominant factor when analyzing the channel system in Changjiang estuary (as outlined in lines 139-144). The new text reads:

“In this context, the primary morphodynamic process at play is the persistent alongshore tidal current within the estuary, which gives rise to steep escarpments along the channel banks. Through the cyclical influence of tides, the formation of parallel channels begins at the escarpment knickpoint (highlighted by a brown box in Fig. 2c), with channels maintaining a perpendicular alignment to the main parent channel (as illustrated in Fig. 2a).”

4. The 80-100-degree demarcation between parallel and oblique channels seems somewhat arbitrary. Is there a justification? It seems that the more important counterfactual against which to test the mechanism for the development of parallel channels would be the development of sinuous channels. There is a strong precedent in fluvial geomorphology for contrasting processes that produce straight versus sinuous

channels.

Reply: We agree with the reviewer that using a plus/minus 10 degrees from perfectly perpendicular channels (90 degrees) to distinguish from oblique channels is somehow arbitrary. Still, there is no quantitative accepted standard to separate the two groups, since the channel angle has a large variability. In the revised manuscript, we have removed the classification of perpendicular or oblique parallel channels, focusing on some special parallel channel systems observed in remote sensing images and characterized by a regular, nearly perfectly perpendicular morphology. To emphasize the characteristics of the angles of parallel channels, we added a new plot (Fig. 2e) to show the probability distribution of the median of channel angles. Figure 2 in the main text has been modified accordingly.

Fig. 2 The channel systems formed in two typical tidal flat environments with the definition of channel angles marked (a and b) and their cross-sectional profiles (c). Comparisons between connecting channel angles (α) and overall channel angles (β) of natural parallel channel systems (d). The probability distribution of the median of channel angles for 21 selected regions (e). S_α and S_β express the skewness of the distribution, which is a measure of the asymmetry of the data around the sample mean, with the positive value signifying the data spreads out more to the right of the mean than to the left, while the negative value signifying the data spreads out more to the left. K_α and K_β express the kurtosis of the distribution, which is a measure of how outlier-prone a distribution is.

The modified sentences now read (lines 100-118):

“The overall distribution of channel angles and the probability distribution of the median of the angles for all the 21 selected regions are shown in Fig. 2d-e. Different from distributary channel networks found in river deltas, whose mean bifurcation angle is typically close to 72° ²², our analysis reveals that the mean intersection angle, α , and mean trend angle, β , of all the selected channel systems are 78.9° and 83.9° , respectively. The variability in channel angles, with a large portion of empirical data showcasing values smaller than 90° (Fig. 2e), suggests a potential deflection of near-shore flows toward the dominant along-shore flow direction, whether it be ebb or flood. Whereas some parallel tidal channel systems show regular patterns with angles approaching 90° (i.e., Fig. 1a-f). The higher degree of morphological variability we observed across our study sites likely results from local variations in tidal hydrodynamics, landforms, vegetation distribution, and combinations thereof. For instance, both the Colorado River Delta in Baja California, Mexico, and Thengar Island in Bangladesh are situated within estuaries. This location aids in determining flood directions from satellite images by examining the estuary's shape. Within these settings (refer to Fig. 1i, j), parallel channel systems are shaped by alongshore tidal currents originating from two distinct parent channels, leading to generally smaller channel angles overall. Notably though, even in the case of parallel channels, variance in the intersection angles is more pronounced than in the channel trend angles (Fig. 2d), and the probability distribution of the median of intersection angles has a smaller peakedness and more significant skewed shape (Fig. 2e), again suggesting larger morphological variability near the shoreline.”

The modified text in lines 129-132 reads: “As depicted in Fig. 1a-f, parallel channel systems exhibiting regular patterns emerge on intertidal flat areas bordering parent channels characterized by steep banks. We thus contend that tidal flat topography serves as the principal contributor to the observed variability in channel angles.”

The modified text in lines 149-151: “These observations highlight the complex interplay of ecomorphodynamic processes that contribute to the morphological diversity observed even within apparently similar parallel tidal channel systems.”

The modified text in lines 157-158: “The above-mentioned observations suggest that the topographic profile of channel banks plays a critical role in dictating the pattern of parallel channel systems.”

We also agree with the reviewers regarding the distinction between straight and meandering channel mechanisms. In the field of river studies, it has been observed that channels in areas with steep slopes are typically straight, whereas those in gently sloping regions exhibit more meandering patterns, as shown in Figure S3a. *Lazarus and Constantine (2013)* attempted to provide a general explanation, suggesting that a higher

Froude number reduces the degree of meandering. The Froude number represents the ratio of hydraulic slope to flow resistance. Lower Froude numbers imply increased friction, resulting in greater resistance to channel flow. Under these conditions, the system, striving to enhance drainage efficiency, opts for increased sinuosity to extend the channel length.

Figure S3. (a) The meandering river on the flat terrain (marked by the red dashed line) and the straight gullies on the cliffs on both sides (marked in a blue box) show the control of the hydraulic slope over the channel shape in the terrestrial environment (modified from *Lazarus and Constantine, 2013*). (b) The parent channel and the parallel channel systems in Persian Gulf, Shatt al-Arab River Mouth, Iran (Source: Google Earth).

This explanation extends to the tidal environments over mudflats, as discussed in our previously published work (*Finotello et. al., 2018*). As shown in Figure S3b, the regular pattern of parallel channels does not extend far into the interior of the mudflat. This is because in the region adjacent to the main channel, where the flow is strong, the flow dynamics is similar to mountainous areas with a higher Froude number. On the other hand, the interior of mudflat is more consistent with the friction-dominant hypothesis proposed by *Rinaldo et. al. (1999)*, and therefore exhibits a more meander pattern.

To further visualize this phenomenon, we compared the morphology of the tidal channel systems in Venice Lagoon, Italy and in Mokpo, west coast of South Korea (Figure S4). The Venice lagoon, characterized by its small tidal range and significant vegetation, demonstrates a clear friction dominance, resulting in a meandering channel morphology. Conversely, in the Korean region, where the tidal range is substantial, especially along the large tidal channels, the friction component is relatively lower, leading to the pronounced development of straight, parallel channels.

The above analysis has been reported in Section 1 of Supplementary Material. We acknowledge the valuable point raised by the reviewer and have accordingly added

some text to elaborate on this point (in lines 46-57):

“Decades of research have been devoted to dissecting the intricate morphologies that define these networks, with particular emphasis on their chaotic branching patterns and meandering extensions⁷⁻²²..... Even though the channel branches in different systems exhibit distinct degrees of curvature, which is dominated by local flow strength and bed friction^{8,9} (also introduced in Section 1 of the supplemental material file), the shape and trend of each branch in the same parallel tidal channel system are consistent. This exceptional uniformity has prompted some researchers to delineate them as a separate entity, aptly termed the Parallel Tidal Channel System.”

Figure S4. (a) The meandering tidal channels observed in the Venice Lagoon, characterized by small tidal ranges and vegetation. (b&c) Straight tidal channels observed on the west coast of Korea, characterized by strong tides and absence of vegetation. The difference between them highlights the controlling effects of friction on channel pattern.

On the other hand, we tend to be brief about the meander patterns, as it is not the central theme of this paper. This decision aligns with the reviewer's comment that “there is a strong precedent in fluvial geomorphology contrasting the processes producing straight versus sinuous channels”, which has been extensively explored in previous literature, including our own publications. We are currently developing a model tailored to the complexities of tidal environments, focusing on meander and branching patterns (see the movie in “Simulated channel meandering.pptx”), to unearth new insights into

this issue. Detailed discussions on the meander characteristics of parallel channel systems are planned for future work. The paper concludes with forward-looking statements reflecting this intent (in lines 312-314):

“Moreover, detailed investigations are needed to examine aspects such as the characteristic spacing of channel branches, drainage efficiency, sinuosity development over time, and possible channel bifurcations.”

Lazarus, E. D., & Constantine, J. A. (2013). Generic theory for channel sinuosity. Proceedings of the National Academy of Sciences, 110(21), 8447-8452.

Finotello, A., Lanzoni, S., Ghinassi, M., Marani, M., Rinaldo, A., & D’Alpaos, A. (2018). Field migration rates of tidal meanders recapitulate fluvial morphodynamics. Proceedings of the National Academy of Sciences, 115(7), 1463-1468.

Rinaldo, A., Fagherazzi, S., Lanzoni, S., Marani, M. & Dietrich, W. E. Tidal networks: 3. Landscape-forming discharges and studies in empirical geomorphic relationships. Water Resources Research 35, 3919-3929 (1999).

5. The connection between the remote sensing component of the project and the modeling was a bit weak, and it seems that opportunities to test the hypothesis through observational data were missed (see comments below on Recommendations for Future Work).

Reply: Agreed. In the previous manuscript, we only qualitatively reproduced the pattern and evolution of the real parallel channel system through numerical modelling. To improve the comparison of the channel patterns between the model and the reality, we have further provided additional observational evidences (such as channel angle, width, length, and spacing) to support our findings.

First of all, as we mainly focus on the channel angle in this study, we analyzed the consistency of the channel angle between the model and the remote sensing image. We found that the overall trend angle of the parallel channels in both the model and the reality is more inclined to 90°, while the intersection angle is smaller and more varied. The modified discussion now reads (in lines 239-246):

“Consistently with the channel systems observed through remote sensing (Fig. 2e), all the simulated channels are perpendicular to the parent channel, with trend angles (β) being approximately equal 90°. It indicates that the alongshore currents turn to the cross-shore direction when traveling in shallow waters, no matter what the tidal environment is. In contrast, a smaller value and a higher variability are observed for the intersection angles (α) as shown in Fig. 4d, which agree with the more dispersed distribution of the median of intersection angle in remote sensing statistics (Fig. 2e), indicating that the pattern of the parallel branch root is more easily affected by the alongshore currents.”

Secondly, we compared the plane morphology of channel systems between the model and the real situation, showing in Fig. S5&S6.

Figure. S5. (a) The mean channel spacings of the 20 selected areas and the model results are plotted as a function of mean channel width. (b) The channel spacings of 3 areas (Hooge Plates, Netherlands, Warbah Island, Kuwait, and Shoalwater Bay, Australia) are plotted as a function of channel width.

Figure. S6. The mean channel length of the 20 selected areas and model results are plotted as a function of mean channel width (a). The relative channel length (calculated by the ratio of

mean channel length and mean channel spacing) are plotted as a function of relative channel width (calculated by the ratio of mean channel width and mean channel spacing).

Note that Fig. S5 - S6 do not include the situation of Jiangsu Coast, China, because the channel system in Jiangsu Coast is too large compared with other systems.

The mean channel spacing and mean channel width of the field channel systems approximately follow a linear relation (Fig. S5a). However, the model results do not agree with the trend of the tidal channels in the field, and the mean channel width is relatively larger. In Fig. S5b, c_x and c_y are the dispersion coefficient of channel width and channel spacing, respectively. It shows that larger tidal channel systems also have greater degree of variation.

Due to the large grid in the model, the simulated tidal gully has a large scale, while the mean channel length and the mean channel width in both model and reality shows a linear increase trend (Fig. S6a). Fig. S6b shows the relative channel length as a function of relative channel width. Divided by the mean channel spacing, the two parameters express the degree of channelization per unit tidal flat area. Because of the distinct environmental conditions in different areas, the field data distributes dispersedly in Fig. S6b. Moreover, the model results show a larger trend than the field data, indicating that the simulated condition in the model tends to generate denser channel system.

These comparisons are also reported in the Section 4 of the Supplementary Material. In the main text, we cited this content in lines 178-180: “The model successfully captures the formation and evolution of parallel channel systems formed by alongshore tidal flows (detailed comparisons of the channel patterns between the model and the reality can be found in Section 4 of the supplementary material).”

On the other hand, to further explain the contribution of the alongshore currents and verify the reasonability of the simulation, we compared the model and the reality in terms of velocity field. We found on-site data showing that steep escarpments may also result in a more rapid change in flow direction compared to linearly-sloping bed conditions. As observed by *Zhang et. al. (2016&2021)* in the Jiangsu Coast (shown in Fig. S7), the alongshore component of the velocity ellipse at the seaward station A, is much larger than the cross-shore one, indicating that the seaward area is dominated by the alongshore currents. Moving landward to stations S7 and S6, the cross-shore velocity component gradually increases, which is consistent with to the model results for the case of linearly-sloping bed.

Figure S7. On-site observations of flow field in Jiangsu Coast (modified from Zhang *et al.*, 2016&2021).

Fig. S8 portrays the flow field measured in the field (Dai *et al.*, 2013) in Chongming island, whose bed profile is characterized by a sharp transition in elevation at the bank of the north branch of Changjiang river. In the north branch of Changjiang river, the flow direction is mainly along the channel, and the transversal velocity is restricted by the channel bank, consistent with the channel velocity in the simulated scarp-shaped case (see Fig. 3a in the main text).

Figure S8. Spatial variations in flood/ebb current directions in the North Branch and South Branch (Synchronous survey date: 22 Sept. 2002) The location of the parallel channel system in our study is marked by the red star (modified from *Dai et. al., 2013*).

The comparison with field observations has been added in Section 3 of Supplementary Material and in lines 300-308 of the main text:

“Field observations also documented the process of changing flow direction as the tide spreads between the tidal channel and the tidal platform³¹⁻³³ (see Fig. S3.2 and Fig. S3.3 in Section 3 of Supplementary Material). With the gradual rise of the water level, there is a transition of flow direction from alongshore to cross-shore. Furthermore, field observations carried out in the gently-sloping tidal flat on the Jiangsu Coast (China), indicate that the alongshore component of the velocity ellipse gradually decreases moving landward, while the cross-shore component increases^{33,34}. Conversely, in the north branch of Changjiang River, the velocity is mainly oriented along the channel, due to the flow restriction imposed by the steep channel banks³⁵.”

Zhang, Q. et al. The role of surges during periods of very shallow water on sediment transport over tidal flats. Frontiers in Marine Science 8, 599799 (2021).

Zhang, Q. et al. Velocity and sediment surge: What do we see at times of very shallow water on intertidal mudflats? CONTINENTAL SHELF RESEARCH 113, 10-20 (2016).

Dai, Z., Fagherazzi, S., Mei, X., Chen, J., & Meng, Y. (2016). Linking the infilling of the North Branch in the Changjiang (Yangtze) estuary to anthropogenic activities from 1958 to 2013. Marine Geology, 379, 1-12.

6. There is no statement of data/code availability, and there is not enough data/information for the work to be reproduced.

Reply: Agreed. We have incorporated sections for data and code availability, providing detailed information on the data concerning worldwide parallel channels that we collected, as well as the model source code and setup. This ensures full reproducibility of the study. The added text is provided below:

Data availability:

The data regarding the worldwide parallel channels are available as supplementary material (Supplementary Data.xlsx).

Code availability:

The model Delft3D is an open-source code available online (at <https://oss.deltares.nl>). The model set-up is available at the repository Zenodo <https://zenodo.org/records/10814687>)

MODERATE AND MINOR COMMENTS

- The model is mentioned only very briefly outside the supplemental section. I suggest including at least a paragraph describing the model in the main text, as details about what the model does and does not include are essential to the interpretation of the results.

Reply: Agreed. We have added a model description in the main text to clarify the model used in this study. The added text reads (in lines 170-177):

“Numerical simulations were conducted utilizing Delft3D, a hydro-morphodynamic modeling package that employs finite difference methods to solve depth-averaged shallow water equations, facilitating the simulation of water levels and flow velocities (further details about the model can be found in Section 3 of the supplementary material). Subsequently, the hydrodynamic outputs were utilized to estimate sediment transport of both cohesive and non-cohesive sediment fluxes. The resulting morphological alterations were computed and updated at each hydrodynamic time step. The updated bed level was then employed to calculate the flow field at the subsequent hydrodynamic time step, thereby completing the morphodynamic loop^{13,29}.”

- Even in Supplementary Section 3, the model description does not contain an adequate level of detail, including description of how the equations were solved (e.g., via finite element or finite difference) and on what platform, what the grid resolution and time step were, and what the boundary conditions were for water and sediment. Is the code openly available to readers? I would encourage the authors to embrace best practices for open, reproducible science in making data and code available.

Reply: Agreed. We have elaborated the model description in the Supplementary Section 3.

- Line 104-105: “data dispersion of intersection angles is more pronounced than in the trend angles.” I had to read this phrase many times before it made sense to me. Please

rephrase. Maybe “Variance in the intersection angles is more pronounced than in the channel trend angles.”

Reply: We have replaced the text following the suggestion of the reviewer. Additionally, we added some description about the new plot in Fig. 2e. It now reads (in lines 115-118):

“Notably though, even in the case of parallel channels, variance in the intersection angles is more pronounced than in the channel trend angles (Fig. 2d), and the probability distribution of the median of intersection angles has a smaller peakedness and more significant skewed shape (Fig. 2e), again suggesting larger morphological variability near the shoreline.”

• Figure 3: The subfigures are mislabeled in the caption and in the text references.

Reply: Thanks. This has been fixed. The revised Fig. 3 caption reads:

“**Fig. 3.** Numerical modeling of parallel tidal channel systems: The initial bed elevation and velocity variations of the two numerical models (a-b). The modeled tidal channel systems after five years (c-d). The two initial bed profiles and the mean flow trend angle of the two cases (e). The blue and yellow vertical dashed lines denote the lower boundaries of the intertidal zone of the two cases.”

Line 178: Larger compared to what?

Reply: The variation processes of the flow trend angle and overall channel angle from sea to land are described in this sentence (as illustrated in Fig. 3 in the main text). In the inner area, where water depth is shallower, the current flow shifts to a cross-shore orientation, resulting in larger angles in the inner area compared to the seaward area. The sentence has been modified as follows (lines 213-215):

“Due to the continuity of tidal currents, the alongshore flow shifts to cross-shore to feed and drain the inner area, leading to landward-increasing values of flow trend angle (γ) and overall channel angle (β).”

• Line 224-226: This statement appears to be driven by one result. I’m not convinced that it is an accurate/representative statement.

Reply: Thanks for your suggestion. Indeed, the correlation between the channel angle and the channel size is somewhat weak. These statements are based on the angle statistics (shown in Fig. 4d in the main text) and channel morphology analysis (shown in Fig. 4c in the main text). We have modified this sentence, which now reads (lines 266-268):

“On the one hand, increased local topographic relief leads to a greater spread in the overall angles of the tidal channels. The variable topography tends to generate tidal channel systems of varying sizes and uneven distribution (Fig. 4c).”

- Figure 4a: I suggest omitting the TR in the legend of Figure 4a, as the abbreviation is not defined and it is not consistent with other subfigures.

Reply: Thanks for your suggestion. Fig. 4 has been modified accordingly.

- Paragraph starting on line 261: Much of this text is repetitive of the first paragraph in this section.

Reply: Agreed. We have merged this paragraph with the preceding paragraph to avoid repetition. Additionally, we added a new paragraph at the end of discussion part to illustrate the implications of this study for coastal management and ecological conservation. It reads (in lines 317-326):

“The systematic study of parallel tidal trench system can provide strong support for coastal management and ecological environment protection in the future. First of all, since the parallel tidal channel system is controlled by hydrodynamic and topographic conditions, managers can judge the local environmental conditions and changes based on the morphology of parallel channels through remote sensing, such as the tidal current regime and ecosystem states³⁶, providing help for the overall management of tidal flats, especially in the context of global climate change such as sea level rise. Secondly, it also provides a way to artificially shape the tidal channel system: by adjusting the strength of the alongshore current and the surrounding terrain, the secondary tidal trench system can be shaped as needed to meet the local demand for water, sand, and material exchanges.”

- Line 282-284: The statement about relevance to management is vague. I suggest deleting it or making it more specific.

Reply: Agreed. Following your previous suggestion, we briefly described the possible contribution of this study to coastal environmental management. Here, we modified this sentence. It now reads (in lines 339-342):

“This work provides valuable insights into the factors controlling observed morphological variability in channel networks around the world. By reasonably predicting the development trend of tidal channels, it can contribute to the management and preservation of tidal environments.”

RECOMMENDATIONS FOR FUTURE WORK

If the authors are looking to extend their analysis in their future work, it would be interesting to see whether varying spatial features of an imposed topography (such as a wavelength, if the initial topography were to be initialized nonrandomly) would influence the outcome, given that multipatch flow-sediment feedbacks are strongly dependent on the spacing of roughness elements.

Another interesting element to consider in future work might be extracting digital

elevation model data from the marshes analyzed through remote sensing. A powerful test of the authors' hypothesis might be whether the topographic characteristics associated with parallel vs. oblique channels in the observational data agree with the conclusions from modeling.

Reply: Thanks for the insightful recommendations. As you suggested, we aim to generate scholarly interest in parallel tidal channel systems through this study, promoting further investigation into their distinct characteristics, including drainage density and spatial distribution attributes (such as the spacing of channels). Investigating the developmental trends and morphological features of tidal channels under artificially-imposed topography is a compelling and significant area of inquiry. This approach offers a natural means to alter tidal flat topography and the surrounding environment, which holds particular relevance in achieving a balance between economic development and environmental preservation. We have included a paragraph in the discussion section to delineate potential avenues for future research (referenced in lines 309-316):

“Further in-depth and systematic research is needed to enhance our understanding of parallel tidal channel systems. Numerous factors, such as vegetation growth rates and colonization strategies, sediment physical properties, and human-induced changes in topography, may influence channel morphology. Moreover, detailed investigations are needed to examine aspects such as the characteristic spacing of channel branches, drainage efficiency, sinuosity development over time, and possible channel bifurcations. To advance research in this field, it is essential to collect more comprehensive field data, which should encompass tidal flows, sediment properties, and high-resolution digital elevation data.”

REFERENCES CITED

- Larsen, L. G. Multiscale flow-vegetation-sediment feedbacks in low-gradient landscapes. *Geomorphology* 334, 165–193 (2019).
- Moffett, K. et al. Multiple Stable States and Catastrophic Shifts in Coastal Wetlands: Progress, Challenges, and Opportunities in Validating Theory Using Remote Sensing and Other Methods. *Remote Sensing* 7, 10184–10226 (2015).
- Moffett, K. B. & Gorelick, S. M. Alternative stable states of tidal marsh vegetation patterns and channel complexity. *Ecohydrology* 9, 1639–1662 (2016).

Reviewer #2 (Remarks to the Author):

This is an interesting paper dealing with connectivity geometry between secondary channels and the adjacent water body (trunk channel, bays, open water, etc.) in estuarine environments. They provide examples from around the world to demonstrate the applicability of their study. As stated their major objective was to “disentangle the influence of alongshore tidal currents and bed topography” in the formation of these systems.

Reply: We thank the Reviewer for his/her positive feedback and recommendation.

Figure 2 is difficult to read and understand and is of poor quality. The aerials showing the channels are small and difficult to see. Panel 2d is hard read too. The aerials I Figure 1, c, h, & j are of poor quality as well.

Reply: We initially submitted high-resolution figures that, unfortunately, may have experienced a loss in resolution due to the compression process implemented by the online submission systems. As a result, some details (such as small channels) may not be very clear. To address this concern, we have made slight adjustments to both Figure 1 and Figure 2 to enhance their readability (refer to the modified figures below). Additionally, we will provide the high-resolution figures directly to the editorial department of the journal.

Fig. 1 Global distribution of parallel channel systems with a few examples. Subplots (a-f) are

examples of linear parallel branches that are nearly straight and generally perpendicular to the parent channel; subplots (g-j) are examples of parallel branches that are not very straight and are oblique to the parent channel. Source: Google Earth.

Fig. 2 The channel systems formed in two typical tidal flat environments with the definition of channel angles marked (a and b) and their cross-sectional profiles (c). Comparisons between connecting channel angles (α) and overall channel angles (β) of natural parallel channel systems (d). The probability distribution of the median of channel angles for 21 selected regions (e). S_α and S_β express the skewness of the distribution, which is a measure of the asymmetry of the data around the sample mean, with the positive value signifying the data spreads out more to the right of the mean than to the left, while the negative value signifying the data spreads out more to the left. K_α and K_β express the kurtosis of the distribution, which is a measure of how outlier-prone a distribution is.

Line 102: “For instance, parallel channels in the Colorado River Delta (Baja California, Mexico) and Thengar Island (Bangladesh) are influenced by alongshore tidal currents from two different parent channels, resulting in overall smaller channel angles (see Fig.1i, j).” As this is a remote sensing study, how are authors able to characterize the tidal flow? There are no citations. On what basis are they able to make that observation?

Reply: The direction of water flow can be inferred based on the alignment of large tidal channels in remote sensing images and from numerical simulations conducted in these

areas, as depicted in Fig. S9. Both sites are situated within estuaries. The numerical simulations conducted for the Colorado River Delta (*Montaño et al., 2008*) and Thengar Island (*Jakobsen et al., 2002*) provide insights into the flood direction within the two estuaries, as illustrated in Fig. S9a,e. Consequently, assessing the current direction based on the estuary's morphology becomes straightforward.

Figure S9. Numerical simulations about flood direction in Colorado River Delta and Thengar Island (a & e, modified from *Montaño et al., 2008* and *Jakobsen et al., 2002*). Remote sensing images of parallel channels in Colorado River Delta, Baja California, Mexico (b-d), and Thengar Island, Bangladesh (f-h). Source: Google Earth.

In the Colorado River Delta (Fig. S9b-d), the flood currents would go upward through the main channel, and then bifurcate into the 1st-order branch on the left (Fig. S9d). The width of the 1st-order branch is about 65 m (measured in the remote sensing image, Fig. S9d, which is at a low water level), which is large enough to characterize the flow direction in it.

The Thengar Island (Fig. S9f-h) is situated outside the estuary, and is isolated by two main channels (Fig. S9g-h). Therefore, flood currents on the island originate from two directions. It is important to note that the remote sensing image in Fig. S9h was captured at a low water level, and hence the left main channel is not clearly visible.

In order to clarify these findings, we added a brief explanation to further support the conclusion (in lines 110-115). The new text reads:

“For instance, both the Colorado River Delta in Baja California, Mexico, and Thengar Island in Bangladesh are situated within estuaries. This location aids in determining flood directions from satellite images by examining the estuary's shape. Within these settings (refer to Fig. 1i, j), parallel channel systems are shaped by alongshore tidal currents originating from two distinct parent channels, leading to

generally smaller channel angles overall."

Montaño, Y., & Carbajal, N. (2008). Numerical experiments on the long-term morphodynamics of the Colorado River Delta. Ocean Dynamics, 58, 19-29.

Jakobsen, F., Azam, M. H., & Mahboob-Ul-Kabir, M. (2002). Residual flow in the Meghna Estuary on the coastline of Bangladesh. Estuarine, Coastal and Shelf Science, 55(4), 587-597.

Line 111: "By comparing the topography of tidal flats from the satellite images (Fig. 1), we deem that the variations in tidal flat topography are the main causes for the formation of two types of parallel channels." It has been shown that the erodibility of the substrate is an important parameter in affecting channel evolution, and you present no data on the sedimentology (grain size, shear strength, etc.) of these systems. Moreover, the comparisons between the channel profiles in Figure 2 are confusing, as the Jiangsu channel also appears to have a knickpoint but less dramatic.

Reply: We acknowledge that various environmental factors, including sediment physical properties, can influence the morphological characteristics of tidal channel systems. Additional information regarding the sedimentology of the selected study areas has been provided in Section 2 of the Supplementary Material. However, despite the diverse environmental influences on parallel channel systems worldwide, we have observed that some systems with regular patterns occur alongside parent channels characterized by a sharp transition in elevation. Therefore, we contend that terrain slope is the primary driver for the development and occurrence of parallel channels.

Following the reviewer's comments, we have revised the sentence to provide a clear explanation for our assumption. The updated text now reads (in lines 129-132):

"As depicted in Fig. 1a-f, parallel channel systems exhibiting regular patterns emerge on intertidal flat areas bordering parent channels characterized by steep banks. We thus contend that tidal flat topography serves as the principal contributor to the observed variability in channel angles."

Concerning the comparison between the channel profiles in Fig. 2a-b, the Jiangsu coast also appears to have a knickpoint. However, the most significant difference between the two field sites is the elevation of the knickpoint relative to the mean sea level. In Chongming Island, due to the bank erosion induced by the alongshore tidal currents in the estuary, the escarpment is higher and steeper, generating a sharp change in the topographic profile. On the contrary, the knickpoint on the Jiangsu coast is much lower and slope changes are less pronounced. Therefore, the bed profile comprised within the intertidal range is more uniform and changes more gradually compared with Chongming Island. The difference in topographic profiles will lead to significant distinctions in channel patterns.

To emphasize the key difference between the two cases, we modified the original sentences (in lines 139-147), which now read:

“In this context, the primary morphodynamic process at play is the persistent alongshore tidal current within the estuary, which gives rise to steep escarpments along the channel banks. Through the cyclical influence of tides, the formation of parallel channels begins at the escarpment knickpoint (highlighted by a brown box in Fig. 2c), with channels maintaining a perpendicular alignment to the main parent channel (as illustrated in Fig. 2a). However, in contrast to the sharp change in bed topography observed in Chongming Island, the selected tidal basin in the Jiangsu coast (China) features an approximately uniform seaward bed slope. This characteristic results in a much wider intertidal zone, as illustrated by the blue line in Fig. 2c, and leads to distinct parallel channels”

Line 181. This statement is highly significant because it their explanation of why the two systems are different: “On the contrary, because of the sharp change of bed slope in the Scarped-shaped case, the steep bank restricts the x-directional component of the tidal current when the water level is lower than the inflection point. The tidal currents then experience a fast diversion as they flow over the inflection point on the profile, leading to a relatively large channel angle at the roots of the parallel branches.” However, this needs to be much better stated. In the linear case, the tide gradually reverses flow (from ebb to flood) as the tide rises, because of the gradual slope. However, in the scarped case water doesn’t enter the secondary channels until later in the tidal cycle when the tide is rising steeply thereby producing a rapid change in current direction and strong flood currents. I would also guess that the longshore current reaches a maximum after the flow significantly diminishes in the secondary channel in the linear case. In the scarped case, the longshore current is maximum after the water level has dropped below the secondary channel opening. Again, in paper, this is not well explained.

Reply: Thanks for your suggestions, which provide a valuable perspective on the effects of different bed profiles. Our manuscript primarily delves into the flow dynamics during ebb or flood tides. As highlighted by the reviewer, the timing of these tidal phases is also significant. Our focus lies in examining the flow field within the secondary channel, as it plays a direct role in shaping its development. Thus, we placed particular emphasis on the rate of change in water level when the flow direction shifts within these secondary channels. We have modified this discussion (lines 215-223). It now reads:

“In the Linear-shaped case (blue cross symbols in Fig. 3a), the intertidal zone is wide enough for the alongshore ebb and/or flood currents to adjust their direction. Additionally, as the ebb currents gradually turn to flood currents at low tides, the flow smoothly transitions between 0° and 90°. Conversely, in the Scarped-shaped case, the

sharp change in bed slope results in a steep bank that restricts the x-directional component of the tidal current when the water level is lower than the inflection point. Consequently, tidal currents experience a fast diversion as they flow over the inflection point on the profile, whose elevation is near the mean sea level. This diversion in flow direction occurs quickly because of the steep rise in the tide at this moment, leading to a relatively large channel angle at the roots of the parallel branches.”

The strength and timing of the longshore current is highly germane to this paper, and we are provided with no information of its character (time-velocity asymmetry).

Reply: We have introduced the character of longshore currents in lines 66-71:

“However, the geomorphology and dynamics of parallel channel systems in coastal environments are markedly different: they are normally found in low-gradient landscapes such as mudflats and marsh surfaces^{4,25}, where the erosion processes are largely controlled by water-surface gradients rather than topographic ones^{8,9,11,26}, and intriguingly, their dynamics are dominated by alongshore tidal flows, whose strength and water level change periodically, yet their orientation is cross-shore.”

Additionally, in the discussion part “Condition for the formation of parallel channel systems”, we added the qualitative comparison between the model and the field measurement to illustrate the change process of the flow direction of the longshore current in the tidal cycle and prove the rationality of the model simulation (lines 300-308). It now reads:

“Field observations also documented the process of changing flow direction as the tide spreads between the tidal channel and the tidal platform³¹⁻³³ (see Fig. S3.2 and Fig. S3.3 in Section 3 of Supplementary Material). With the gradual rise of the water level, there is a transition of flow direction from alongshore to cross-shore. Furthermore, field observations carried out in the gently-sloping tidal flat on the Jiangsu Coast (China), indicate that the alongshore component of the velocity ellipse gradually decreases moving landward, while the cross-shore component increases^{33,34}. Conversely, in the north branch of Changjiang River, the velocity is mainly oriented along the channel, due to the flow restriction imposed by the steep channel banks³⁵.”

Larger tidal ranges lead to greater tidal prisms and flow of the longshore current thereby increasing the tendency of longshore transport, spit formation and deflection of the secondary channel mouth.

A major deficiency of this paper is the lack of discussion of how tidal flats versus marsh systems behave.

Reply: The influence of vegetation on parallel tidal channels presents an intriguing and complex subject that warrants systematic and comprehensive discussion in future studies. In response to the reviewer's suggestion, we have incorporated simulations to

emulate the co-evolution of channels and vegetation, aiming to gain a fundamental understanding of the effects of vegetation. These simulations were conducted using the bio-geomorphological model methods outlined by *Best et al. (2018)*, which account for vegetation growth and its impacts on bed roughness and drag coefficient.

For instance, consider the scenario of the Scarped-shaped case (as illustrated in Fig. S1). After a five-years-long simulation, the presence of vegetation facilitates the elongation of channels, resulting in longer channel branches and increased drainage density (defined by *Marani et al., 2003*). This finding aligns with previous research (e.g., *Kearney et al., 2016; Van de Vijssel et al., 2023; Geng et al., 2023*) suggesting that vegetation increases drainage density and efficiency in intertidal plains dissected by tidal channel networks. However, in cases with vegetation, the structure of the parallel channel system and the channel angle closely resemble those observed in scenarios without vegetation. While this similarity may stem from model limitations, such as its inability to simulate vegetation-driven channel meandering (a feature common to all tidal network morphodynamics models with the exception of *Mariotti and Finotello, 2023*), we believe that the effects of vegetation still require comprehensive exploration and specific analyses that are beyond the aim of the present work.

Figure S1. Comparison of bed topography after one-year and five-years evolution in Scarped-shaped case with and without vegetation.

Additionally, we wish to emphasize that parallel channels can form in coastal environments even in the absence of vegetation. For example, refer to Fig. S2, selected from Fig. 1c,h in the main text, which illustrates this point. This indicates that vegetation presence is not necessary for the formation of parallel channels. In certain coastal areas, parallel channels may form before vegetation colonization occurs, and subsequently established vegetation may strengthen the pre-existing channels, enhancing their stability and promoting second-order planform development such as lateral migration and meandering. However, in situations where vegetation is established prior to or simultaneously with development, the effects of vegetation on parallel channels remain unclear and should be carefully addressed in future studies. We have included a paragraph in the discussion section to elaborate on potential avenues for future research (please see our response to Rev#3 regarding “Recommendations for Future Work”).

Figure S2. Examples of parallel channels observed in bare mud flats, such as in Ems-Dollard, Netherland (a) and in Iranian Bay, Iran (b). These satellite images are also shown in Fig.1 c&h in the main text. Source: Google Earth.

Marani, M., Belluco, E., D'Alpaos, A., Defina, A., Lanzoni, S., & Rinaldo, A. (2003). *On the drainage density of tidal networks. Water Resources Research, 39(2).*

Best, Ü. S., Van der Wegen, M., Dijkstra, J., Willemsen, P. W. J. M., Borsje, B. W., & Roelvink, D. J. (2018). *Do salt marshes survive sea level rise? Modelling wave action, morphodynamics and vegetation dynamics. Environmental modelling & software, 109, 152-166.*

Kearney, W. S., & Fagherazzi, S. (2016). *Salt marsh vegetation promotes efficient tidal channel networks. Nature communications, 7(1), 12287.*

Van de Vijzel, R. C., van Belzen, J., Bouma, T. J., van der Wal, D., Borsje, B. W., Temmerman, S., ... & van de Koppel, J. (2023). *Vegetation controls on channel network complexity in coastal wetlands. Nature communications, 14(1), 7158.*

Geng, L., Lanzoni, S., D'Alpaos, A., Sgarabotto, A., & Gong, Z. (2023). *The Sensitivity of Tidal Channel Systems to Initial Bed Conditions, Vegetation, and Tidal*

Asymmetry. Journal of Geophysical Research: Earth Surface, 128(3), e2022JF006929.

Mariotti, G., & Finotello, A. (2023). A flow-curvature-based model for channel meandering in tidal marshes. ESSOAR Open Archive.

REVIEWERS' COMMENTS

Reviewer #1 (Remarks to the Author):

The authors have done a great job of responding to my comments, and I appreciate the additions in the text and in the Supplementary Materials, which are quite rich. My remaining comments are very minor—editorial and grammatical in nature.

Main text:

- Line 57: Why is *thalweg* italicized? I recommend removing the italics.
- Line 105: The sentence beginning with “whereas” is a fragment. It can be remedied by converting the preceding period to a comma and combining the two sentences.
- Figure 2c: It is odd to have the 0 distance at some arbitrary distance inland. Please consider reversing the x axis so that 0 begins at the channel edge (drawn as the baseline on the figure). This comment also applies to other figures in the Supplementary material.
- Lines 167-174: I appreciate the brief description of the model added to the main text. Some additional additions that would be welcome would be referring to the cohesive sediment fraction as mud and non-cohesive as sand (as in the Supplementary section 3) and mentioning specifically that vegetation is not simulated in the majority of runs.
- Line 184: Should this be 3c and 3d?
- Line 195: Should this be 3c?
- Line 311: trench system \diamond channels
- Line 319: trench \diamond channel
- Line 323: imagery \diamond images
- Line 330: are share \diamond is sharp

Supplementary:

- Line 46: does \diamond do
- Line 47 focuses \diamond focus
- Table S3.1-S3.2: Not all parameters/variables in the model are defined in the table, such as S_{sand} , τ_{max} , s_x , and s_y (I realize these are variables, not parameters, but should be defined somewhere, likely the paragraph beginning on line 191), and A_s .
- Line 266: Change this to “Information about tidal range is acquired from ArcGIS map viewer”
- Line 296: shows \diamond show
- Line 297: increase \diamond increasing
- Section 5: Consider providing more context at the beginning of this section rather than simply “Poisson equation.”

Reviewer #1 (Remarks on code availability):

I clicked on the link to the Delft 3D model setup (a .rar file) to verify it is present. However, I am not equipped to run Delft 3D, so I cannot comment on the reproducibility of the code.

Reply to Reviewers

Manuscript No.: NCOMMS-23-47951-T

Manuscript title: Cross-shore parallel tidal channel systems formed by alongshore currents

Authored by: Zeng Zhou, Yizhang Wei, Liang Geng, Ying Zhang, Yuxian Gu, Alvise Finotello, Andrea D'Alpaos, Zheng Gong, Fan Xu, Changkuan Zhang, Giovanni Coco

Date of initial submission: October 8, 2023

Date of decision email sent: April 26, 2024

Note to the Reviewers:

The comments and suggestions of the reviewers are copied in normal font. The reply to each comment by the reviewers is written in **blue font** and appears just after the original comment or question. The modified/added sentences have been copied from the revised manuscript for the convenience of the reviewers.

We wish to thank the reviewers for their valuable and constructive comments that have certainly resulted in a much more insightful manuscript.

Reviewer Comments

Reviewer #1 (Remarks to the Author):

The authors have done a great job of responding to my comments, and I appreciate the additions in the text and in the Supplementary Materials, which are quite rich. My remaining comments are very minor—editorial and grammatical in nature.

Reply: We wish to thank the reviewer again for a thorough review work, which have been very constructive in our revision of the manuscript. We have complied with all recommendations for change, which are documented in our reply.

Main text:

- Line 57: Why is *thalweg* italicized? I recommend removing the italics.

Reply: Done, as the reviewer suggested.

- Line 105: The sentence beginning with “whereas” is a fragment. It can be remedied by converting the preceding period to a comma and combining the two sentences.

Reply: We have modified this sentence. Now it reads:

“However, some parallel tidal channel systems show regular patterns with angles approaching 90° (i.e., Fig. 1a-f).”

- Figure 2c: It is odd to have the 0 distance at some arbitrary distance inland. Please

consider reversing the x axis so that 0 begins at the channel edge (drawn as the baseline on the figure). This comment also applies to other figures in the Supplementary material.

Reply: Thanks for your suggestion. We have modified the figure accordingly (see below). Additionally, we changed the title of x axis to “Relative distance from deep channels (km)”.

• Lines 167-174: I appreciate the brief description of the model added to the main text.

Some additional additions that would be welcome would be referring to the cohesive sediment fraction as mud and non-cohesive as sand (as in the Supplementary section 3) and mentioning specifically that vegetation is not simulated in the majority of runs.

Reply: Agreed. We have modified this paragraph. Now it reads:

“Numerical simulations were conducted utilizing Delft3D, a hydro-morphodynamic modeling package that employs finite difference methods to solve depth-averaged shallow water equations, facilitating the simulation of water levels and flow velocities (further details about the model can be found in Section 3 of the supplementary material). Subsequently, the hydrodynamic outputs were utilized to estimate sediment transport of both cohesive (i.e., mud) and non-cohesive (i.e., sand) sediment fluxes. The resulting morphological alterations were computed and updated at each hydrodynamic time step. The updated bed level was then employed to calculate the flow field at the subsequent hydrodynamic time step, thereby completing the morphodynamic loop^{13,29}. In this study, we only considered the bare tidal flats, and the effects of vegetation was

not simulated.”

- Line 184: Should this be 3c and 3d?

Reply: Yes. It has been corrected.

- Line 195: Should this be 3c?

Reply: It should be Fig. 3c-d. We have corrected it.

- Line 311: trench system → channels

Reply: Agreed. It has been corrected.

- Line 319: trench → channel

Reply: Agreed. It has been corrected.

- Line 323: imagery → images

Reply: Agreed. It has been corrected.

- Line 330: are share → is sharp

Reply: Agreed. It has been corrected.

Supplementary:

- Line 46: does → do

Reply: Agreed. It has been corrected.

- Line 47 focuses → focus

Reply: Agreed. It has been corrected.

- Table S3.1-S3.2: Not all parameters/variables in the model are defined in the table, such as S_{sand} , τ_{max} , s_x , and s_y (I realize these are variables, not parameters, but should be defined somewhere, likely the paragraph beginning on line 191), and A_s .

Reply: Thanks. We have added a new paragraph to define these variables. It reads: “where \$u\$ and \$v\$ are the depth-averaged velocities in x and y directions (m/s), respectively; \$t\$ is time (s); \$g\$ is the gravitational acceleration (m/s²); \$f\$ is the Coriolis force coefficient (1/s); \$h\$ is the water depth (m); \$\eta\$ is the water level with respect to datum (m); \$\nu\$ is the eddy viscosity coefficient (m²/s); \$C\$ is the Chézy friction coefficient (m^{1/2}/s); \$\tau_{max}\$ is the maximum bed shear stress (Pa); \$Q_{mud,e}\$ and \$Q_{mud,d}\$ are respectively erosion and deposition fluxes described by the widely-adopted Partheniades–Krone formulations; \$M_e\$ is the erosion parameter (kg/m²/s); \$w_s\$ is the settling velocity (m/s); \$c\$ is the depth-averaged concentration (kg/m³); \$\tau_{cr,e}\$ and \$\tau_{cr,d}\$ are the critical shear stress for erosion and deposition of mud fraction,

respectively (Pa); S_{sand} is the total sediment transport of sand particles (m²/s); A_s is the parameter related to sediment properties and water depth; C_{dc} is a non-dimensional drag coefficient due to current alone; U_{rms} is the root-mean-square wave orbital velocity that can be related to wave-orbital speed (m/s); u_{cr} is the threshold velocity for sediment mobilization and is estimated as a function of sediment grain size (m/s); ϵ is bed porosity; z is bed level; S_x and S_y are sediment transports in x and y directions, respectively.”

- Line 266: Change this to “Information about tidal range is acquired from ArcGIS map viewer”

Reply: Agreed. Changed accordingly.

- Line 296: shows → show

Reply: Agreed. Changed accordingly.

- Line 297: increase → increasing

Reply: Agreed. Changed accordingly.

- Section 5: Consider providing more context at the beginning of this section rather than simply “Poisson equation.”

Reply: Thanks for your suggestion. We added a new paragraph to explain the significance of section 5. It reads:

“The characteristic tidal flow field over tidal networks and flats is friction-dominated and can be approximated by a Poisson-type equation deduced from the shallow water equation. Therefore, the bending process of the streamline is subject to the balance between lateral water surface gradient and bed friction resistance, and can be explained by solving Poisson equation. Poisson equation can be expressed as (Di Silvio et al, 2010):”

Reviewer #1 (Remarks on code availability):

I clicked on the link to the Delft 3D model setup (a .rar file) to verify it is present. However, I am not equipped to run Delft 3D, so I cannot comment on the reproducibility of the code.

Reply: Delft3D is an open-source morphodynamic model maintained by Deltares. This model can be downloaded freely via <https://oss.deltares.nl> after registration. The source code is way too heavy to be uploaded, therefore we have suggested the link to download the model and we have also provided the model setup files, so that all the model results presented in this manuscript can be reproduced.